# Preference-Driven Multi-Objective Combinatorial Optimization with Conditional Computation

**Mingfeng Fan**
National University of Singapore
ming.fan@nus.edu.sg

**Jianan Zhou**[*]
Nanyang Technological University
jianan004@e.ntu.edu.sg

**Yifeng Zhang**
National University of Singapore
yifeng@u.nus.edu

**Yaoxin Wu**[*]
Eindhoven University of Technology
y.wu2@tue.nl

**Jinbiao Chen**
Sun Yat-sen University
chenjb69@mail2.sysu.edu.cn

**Guillaume Adrien Sartoretti**
National University of Singapore
guillaume.sartoretti@nus.edu.sg

## Abstract

Recent deep reinforcement learning methods have achieved remarkable success in solving multi-objective combinatorial optimization problems (MOCOPs) by decomposing them into multiple subproblems, each associated with a specific weight vector. However, these methods typically treat all subproblems equally and solve them using a single model, hindering the effective exploration of the solution space and thus leading to suboptimal performance. To overcome the limitation, we propose POCCO, a novel plug-and-play framework that enables adaptive selection of model structures for subproblems, which are subsequently optimized based on preference signals rather than explicit reward values. Specifically, we design a conditional computation block that routes subproblems to specialized neural architectures. Moreover, we propose a preference-driven optimization algorithm that learns pairwise preferences between winning and losing solutions. We evaluate the efficacy and versatility of POCCO by applying it to two state-of-the-art neural methods for MOCOPs. Experimental results across four classic MOCOP benchmarks demonstrate its significant superiority and strong generalization.

## 1 Introduction

Multi-objective combinatorial optimization problems (MOCOPs) involve optimizing multiple conflicting objectives within a discrete decision space. They have attracted considerable attention from the computer science and operations research communities due to their widespread applications in manufacturing [1], logistics [25], and scheduling [17]. In such scenarios, decision makers must simultaneously consider and balance multiple criteria, such as cost, makespan, and environmental impact. Unlike single-objective combinatorial optimization problems (SOCOPs), which seek a single optimal solution, MOCOPs aim to identify Pareto optimal solutions that reflect trade-offs among conflicting objectives, making them inherently more challenging. Given their NP-hard nature, exact methods typically struggle to solve MOCOPs within reasonable time frames, as computational complexity may increase exponentially with problem scale [13, 16]. Consequently, heuristic approaches have emerged as the main avenue for tackling MOCOPs. However, these heuristics often involve extensive iterative

---

[*]Corresponding author.

39th Conference on Neural Information Processing Systems (NeurIPS 2025).

local searches for each new instance, resulting in high computational costs. Furthermore, conventional heuristics often rely on extensive domain-specific expertise and meticulous, problem-specific tuning, thus limiting their adaptability to broader classes of MOCOPs.

Recently, neural methods have achieved great success in solving SOCOPs [3, 8, 22, 27, 33, 43, 60, 69, 70] by learning effective patterns of decision policies in a data-driven way. Motivated by these advances, researchers have extended neural approaches to MOCOPs, leveraging their advantages in bypassing labor-intensive heuristic design, accelerating problem solving through GPU parallelization, and flexibly adapting to diverse MOCOP variants. Typically, neural methods address MOCOPs by decomposing them into a set of scalarized subproblems, each a SOCOP defined by a specific weight vector, and solving them using deep reinforcement learning (DRL) to approximate the Pareto front. Early approaches train or fine-tune a separate model for each subproblem using transfer learning or meta-learning techniques [34, 67]. However, these approaches require extensive computational resources and struggle to generalize to subproblems with unseen weight vectors. As an alternative, PMOCO [38] employs a weight-conditioned hypernetwork to modulate model parameters, allowing a single model to address all subproblems. Nevertheless, it remains limited in effectively handling subproblems with diverse weight vectors. More recently, methods such as CNH [14] and WE-CA [6] have tackled this challenge by encoding weight vectors directly into the problem representations, resulting in a unified model that generalizes across various problem sizes. These methods are currently considered state-of-the-art (SOTA) in solving MOCOPs.

Current SOTA methods typically rely on a single neural network with limited capacity to handle all subproblems, which overcomplicates the learning task and results in suboptimal performance. A straightforward solution to ease training and promote effective representation learning across subproblems is to increase the model capacity. However, determining *how much* additional capacity to allocate and *where* to introduce it within the architecture remains an open challenge. On the other hand, neural methods often adopt REINFORCE [56] as the training algorithm, relying solely on scalarized objective values as reward signals to guide policy updates. Given its on-policy nature, REINFORCE suffers from high gradient variance and lacks structured mechanisms for effective exploration [32]. These issues are exacerbated in MOCOP settings, where the vast combinatorial action space makes efficient exploration particularly difficult, ultimately hindering policy performance.

To address these issues, we propose POCCO (Preference-driven multi-objective combinatorial Optimization with Conditional COmputation), a plug-and-play framework that augments neural MOCOP methods with two complementary mechanisms. First, POCCO introduces a *conditional computation block* into the decoder, where a sparse gating network dynamically routes each sub-problem through either a selected subset of feed-forward (FF) experts or a parameter-free identity (ID) expert. This design enables subproblems to adaptively select computation routes (i.e., model structures) based on their context, efficiently scaling model capacity and facilitating more effective representation learning. Second, POCCO replaces raw scalarized rewards with *pairwise preference learning*. For each subproblem, the policy samples two trajectories, identifies the better one as the winner, and maximizes a Bradley–Terry (BT) likelihood based on the difference in their average log-likelihoods. Such comparative feedback guides the search toward policies that generate increasingly preferred solutions, enabling exploration of the most promising regions of the search space and more efficient convergence to higher-quality solutions. Our contributions are summarized as follows:

- Conceptually, we address two fundamental limitations of existing approaches for solving MOCOPs: limited exploration within the vast solution space and the reliance on a single, capacity-limited model, which can lead to inefficient learning and suboptimal performance.

- Technically, we propose a conditional computation block that dynamically routes subproblems to tailored neural architectures. Additionally, we develop a preference-driven algorithm leveraging implicit rewards derived from pairwise preference signals between winning and losing solutions, modeled using the BT framework.

- Experimentally, we demonstrate the effectiveness and versatility of POCCO on classical MOCOP benchmarks using two SOTA neural methods. Extensive results show that POCCO not only outperforms all baseline methods but also exhibits superior generalization across diverse problem sizes.

## 2 Related Works

**Traditional methods for MOCOP.** MOCOPs are significantly harder to solve than their single-objective counterparts. Classic cases such as the Traveling Salesman Problem (TSP) and Capacitated Vehicle Routing Problem (CVRP) are already NP-hard, and adding multiple objectives only magnifies the difficulty [16]. Exact algorithms quickly become impractical for large instances or for problems that possess a vast Pareto set [18, 59]. Consequently, research has shifted toward heuristic methods that deliver high-quality Pareto fronts within reasonable time limits [25, 64]. Among these, multi-objective evolutionary algorithms (MOEAs) are widely used. They fall into two main categories: dominance-based MOEAs [11, 12, 46] and decomposition-based MOEAs [15, 26, 65]. Despite their popularity, MOEAs typically demand extensive manual design: practitioners must select and tune crossover, mutation, and selection operators along with many hyperparameters [50, 61, 63], and this labor-intensive hand-engineering often limits overall performance.

**Neural methods for MOCOP.** Inspired by the success of DRL in SOCOPs [4], recent research extends neural methods to MOCOPs by solving a series of scalarized SOCOPs. These neural solvers generally follow two paradigms: *one-to-one* and *one-to-many*. The one-to-one paradigm trains or fine-tunes a separate neural model for each scalarized subproblem. Some approaches apply DRL algorithms individually and transfer parameters across models to accelerate convergence [34, 57]. Others adopt architectures such as Pointer Networks [52] and Attention Models [29, 30], and optimize them using evolutionary strategies [47, 66]. To promote generalization across subproblems, meta-learning techniques are introduced [67]. However, this paradigm suffers from high training overhead and the burden of maintaining multiple models. In contrast, the one-to-many paradigm employs a single neural network to handle all subproblems. This includes hypernetwork-based DRL frameworks that condition on weight vectors [35, 38, 58], and conditional neural heuristics that incorporate instance features, weight information, and problem size [6, 14].

**Preference optimization.** Preference optimization methods, such as Direct Preference Optimization (DPO) [44] and Identity Preference Optimization (IPO) [2], have gained significant traction, particularly in large language model (LLM) training, due to their ability to directly optimize human preferences without relying on explicit reward modeling as required in reinforcement learning from human feedback (RLHF). These methods are typically designed for pairwise preference data, where human annotators identify a preferred output over a less preferred one in response to a given prompt. Another line of research explores simpler preference optimization objectives that do not depend on a reference model [21, 62]. Among them, SimPO [41] proposes using the average log-probability of a generated sequence as an implicit reward. This approach aligns more closely with the model's generation process and improves computational and memory efficiency by eliminating the need for a separate reference model. Some studies [37] have explored applying preference optimization to SOCOPs, but its application to MOCOPs has been rarely investigated.

## 3 Preliminary

### 3.1 MOCOP

An MOCOP instance specified by data $\mathcal{G}$ can be formally defined as $\min_{\pi \in \mathcal{X}} F(\pi) = (f_1(\pi), f_2(\pi), \ldots, f_\kappa(\pi))$, where $F$ is the objective vector with $\kappa$ objective functions, $\pi$ is a feasible solution, and $\mathcal{X}$ denotes the feasible solution space. Due to inherent conflicts among objectives, a single solution that simultaneously optimizes all objectives typically does not exist. Instead, Pareto optimal solutions are sought to represent different trade-offs, often guided by weight vectors, among competing objectives. We define the Pareto properties of the solutions as follows.

*Definition 1 (Pareto Dominance). A solution $\pi \in \mathcal{X}$ is said to dominate another solution $\pi' \in \mathcal{X}$ (i.e., $\pi \prec \pi'$) if and only if $f_i(\pi) \leq f_i(\pi')$, $\forall i \in \{1, \cdots, \kappa\}$, and $F(\pi) \neq F(\pi')$.*

*Definition 2 (Pareto Optimality). A solution $\pi^* \in \mathcal{X}$ is Pareto optimal if it is not dominated by any other solution. Accordingly, the Pareto set $\mathcal{P}$ is defined as all Pareto optimal solutions, i.e., $\mathcal{P} = \{\pi^* \in \mathcal{X} | \nexists \pi \in \mathcal{X} : \pi \prec \pi^*\}$. The Pareto front $\mathcal{F}$ is defined as images of Pareto optimal solutions in the objective space, i.e., $\mathcal{F} = \{F(\pi^*) | \pi^* \in \mathcal{P}\}$.*

**Decomposition-based Methods.** Since solving a SOCOP optimally is NP-hard, MOCOPs are significantly more intractable due to the need to identify Pareto optimal solutions, the quantity of which

grows exponentially with the problem size. Therefore, MOEAs are commonly adopted to compute approximate Pareto solutions. Among them, decomposition-based MOEAs (MOEA/Ds) solve a set of subproblems (i.e., SOCOPs) derived from the original MOCOP, a foundation underlying recent DRL-based neural methods. Specifically, the vanilla MOEA/D utilizes decomposition techniques to scalarize an MOCOP into $N$ subproblems with a set of uniformly distributed weight vectors $\{\lambda_1, \lambda_2, \cdots, \lambda_N\}$, each of which satisfies $\lambda_i = (\lambda_i^1, \cdots, \lambda_i^\kappa)^\top, \forall \lambda_i^j \geq 0$ and $\sum_{j=1}^\kappa \lambda_i^j = 1$.

## 3.2 Subproblem Solving

Given a subproblem $(\mathcal{G}, \lambda_i)$, we formulate the solving process as a Markov Decision Process (MDP). An *agent* iteratively takes the current *state* as input (e.g., the instance information and the partially constructed solution), and outputs a probability distribution over the nodes to be selected next. An *action* corresponds to selecting a node, either greedily or by sampling from the predicted probabilities. The *transition* involves appending the selected node to the partial solution. We parameterize the *policy* $p$ by a deep neural network $\theta$, such that the probability of constructing a complete solution $\pi$ is defined as $p_\theta(\pi|\mathcal{G}, \lambda_i) = \prod_{t=1}^T p_\theta(\pi_t|\pi_{<t}, \mathcal{G}, \lambda_i)$, where $T$ is the total number of steps, and $\pi_t$ and $\pi_{<t}$ represent the selected node and partial solution at the $t$-th step, respectively. The *reward* is defined as the negation of the scalarized objective, e.g., $\mathcal{R}(\pi) = -\sum_{j=1}^\kappa \lambda_i^j f_j(\pi)$ with weighted sum decomposition. The policy network is typically optimized using the REINFORCE [56], which maximizes the expected reward $\mathcal{L}(\theta|\mathcal{G}, \lambda_i) = \mathbb{E}_{p_\theta(\pi|\mathcal{G}, \lambda_i)} \mathcal{R}(\pi)$ via the following gradient estimator:

$$\nabla_\theta \mathcal{L}(\theta|\mathcal{G}, \lambda_i) = \mathbb{E}_{p_\theta(\pi|\mathcal{G}, \lambda_i)}[(\mathcal{R}(\pi) - b(\mathcal{G}))\nabla_\theta \log p_\theta(\pi|\mathcal{G}, \lambda_i)], \tag{1}$$

where the baseline function $b(\cdot)$ reduces the gradient variance and stabilizes the training. There are two primary paradigms to extend the above approach to solve MOCOPs. In the *one-to-one* paradigm, methods sequentially train separate models to solve subproblems with a predefined set of weight vectors. However, they often suffer from low training efficiency and generalize poorly to subproblems with unseen weight vectors. In contrast, the *one-to-many* paradigm trains a single model to solve subproblems for arbitrary weight vectors. This is typically achieved by incorporating a subnetwork that transforms each weight vector into model parameters, thus inducing tailored policies for each subproblem. Our POCCO follows the one-to-many paradigm and introduces two key innovations. Specifically, we design a conditional computation block to dynamically route subproblems to different neural architectures, and propose a preference-driven algorithm to train the model effectively.

# 4 Methodology

## 4.1 Overview

POCCO is a learning-based framework that trains a portfolio of policies to solve a set of scalarized subproblems $\{(\mathcal{G}, \lambda_i)\}_{i=1}^N$, obtained by decomposing an MOCOP instance. Instead of forcing a single policy to handle all subproblems, which often yields bland and suboptimal behavior, POCCO promotes specialization: each policy is encouraged to focus on a subset of subproblems, yielding a diverse policy ensemble. Such diversity is known to enhance multi-task optimization [53] by expanding the exploration space and ultimately improving solution quality. Technically, we achieve this diversity by activating different subsets of model parameters through a CCO block, enabling distinct computational paths to emerge for different subproblems. Moreover, POCCO should encourage each policy to thoroughly explore the combinatorial solution space during training for reducing suboptimality. To achieve this, we replace raw rewards with preference signals. For each subproblem $(\mathcal{G}, \lambda_i)$, we construct a set of winning–losing solution pairs $\{(\pi^{w,j}, \pi^{l,j})\}_{j=1}^K$. Training then proceeds by maximizing the likelihood of the winning solutions while minimizing that of the losing ones, following a BT-style objective. This preference-driven training encourages the learned policy to explore the most promising regions of the search space, leading to more efficient convergence toward higher-quality solutions. Notably, POCCO is a generic, plug-and-play framework that can seamlessly involve different neural solvers for MOCOPs. We demonstrate this by augmenting two SOTA methods, CNH [14] and WE-CA [6], in Section 5.

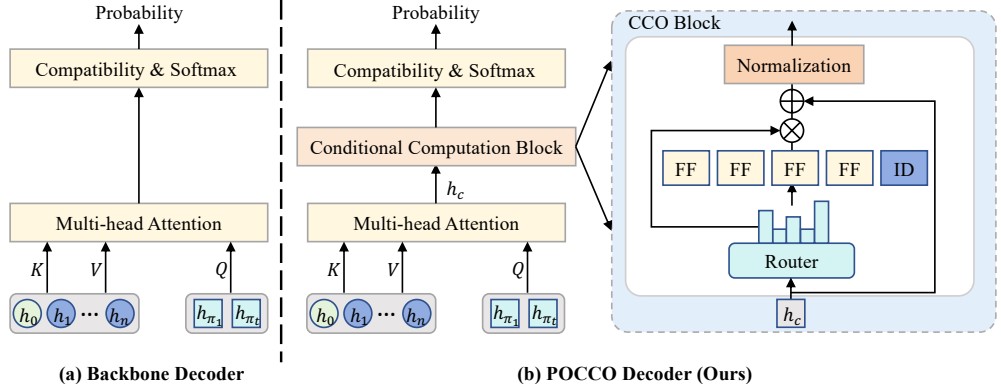

**(a) Backbone Decoder**          **(b) POCCO Decoder (Ours)**

Figure 1: Decoder structures of backbone and POCCO. Given an MOCOP instance $\mathcal{G}$ with $n+1$ nodes (e.g., $n$ customers and a depot, if applicable) and a weight vector $\lambda_i$, POCCO encodes their raw features into joint node embeddings $\{h_i\}_{i=0}^n$ using an encoder. At each decoding step $t$, the decoder forms a query $Q$ from the embeddings of the first and last selected nodes $(\pi_1, \pi_t)$, and computes the key $K$ and value $V$ via linear projections of $\{h_i\}_{i=0}^n$. The MHA layer processes $Q$, $K$, and $V$ to produce a context vector $h_c$, which is refined by the CCO block. The refined context is passed through a compatibility layer followed by a Softmax to compute the node selection probabilities. More details about the forward pass can be found in Appendix C.

## 4.2 Conditional Computational Block

Most approaches employ a Transformer-based architecture, where the encoder generates joint node embeddings $\{h_i\}_{i=0}^n$ that capture the interaction between the instance $\mathcal{G}$ and the weight vector $\lambda_i$, and the decoder produces candidate solutions conditioned on these embeddings. We propose a CCO block to increase the model capacity and promote policy diversity across subproblems. To maintain efficiency, we integrate the CCO block solely into the decoder of the backbone model. This design enables the generation of multiple diverse solutions through a single, computationally expensive encoder pass, offering a favorable trade-off between empirical performance and computational cost.

As illustrated in Fig. 1, the CCO block comprises multiple FF experts and a single ID expert. We insert this block between the multi-head attention (MHA) layer and the compatibility layer in the decoder. Given a batch of MHA outputs $\{h_c^b\}_{b=1}^B$, the CCO block dynamically routes each context vector $h_c^b$ from the corresponding subproblem through either the FF or ID experts, forming distinct computation paths that function as different policies. The ID expert allows the model to bypass the FF computation, promoting architectural sparsity and specialization [19]. Consequently, the CCO block facilitates the learning of dedicated, weight-specific policies tailored to individual subproblems.

Formally, a CCO block consists of: 1) $m$ FF experts $\{E_1, E_2, \ldots, E_m\}$ with independent trainable parameters; 2) a parameter-free identity expert $E_{m+1}$; 3) a router, implemented as a gating network $G$ parameterized by $W_G$, which determines how the inputs $\{h_c^b\}_{b=1}^B$ are routed to the experts; and 4) a skip connection followed by an instance normalization (IN) layer. Given a single context vector $h_c^b$, let $G(h_c^b) \in \mathbb{R}^{m+1}$ denote the output of the gating network, which represents the expert selection probabilities, and let $E_j(h_c^b)$ denote the output of the $j$-th expert. The output of the CCO block is:

$$\text{CCO}(h_c^b) = \text{IN}\left(\sum_{j=1}^{m+1} G(h_c^b)_j E_j(h_c^b) + h_c^b\right). \tag{2}$$

The sparse vector $G(h_c^b)$ activates only a small subset of experts, either parameterized FF experts or the parameter-free ID expert, thereby enabling diverse computation paths while reducing computational overhead. A typical implementation uses a $\text{Top}\,k$ operator that retains the $k$ largest logits and masks the rest with $-\infty$. In this case, the gating network output is: $G(h_c^b) = \text{Softmax}\big(\text{Top}\,k(h_c^b \cdot W_G)\big)$.

Our proposed CCO block aligns with the principles of recent advances in efficiently scaling Transformer-based models along both width [48] and depth [45]. In specific, it combines a mixture-of-experts (MoE) layer, implemented using multiple FF experts to widen the network, with a mixture-of-depths (MoD) layer, realized through an ID expert that allows inputs to skip computation. Within the CCO block, each subproblem is adaptively routed to only a small subset of experts,

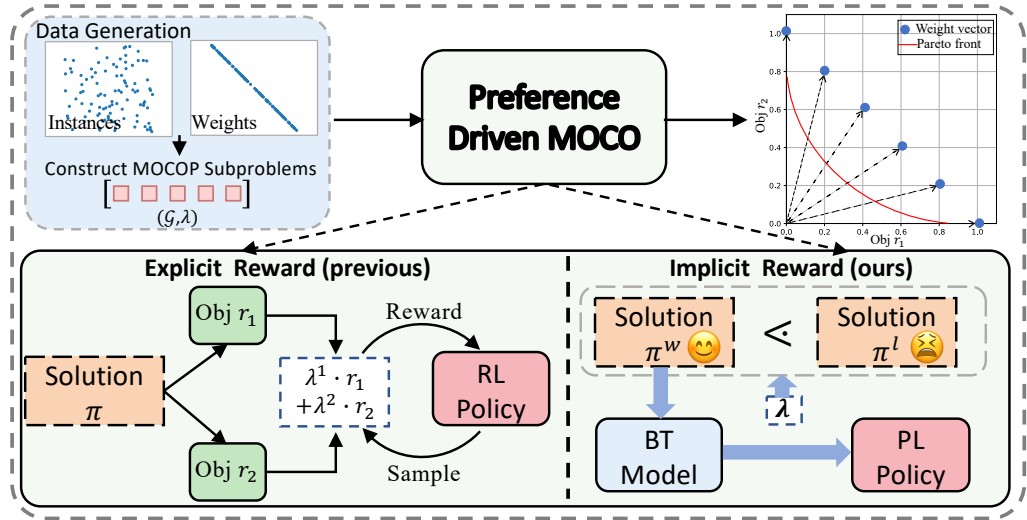

Figure 2: An overview of preference-driven MOCO. Unlike prior DRL methods that explicitly learn from scalarized rewards, our approach converts relative preferences into a BT likelihood, providing an implicit reward signal to optimize the PL policy.

granting the model the expressiveness of a significantly wider network while preserving computational efficiency. As demonstrated in Section 5, this joint design achieves a better capacity–efficiency trade-off than scaling either dimension in isolation.

### 4.3 Preference-driven MOCO

To mitigate the exploration inefficiencies inherent in REINFORCE algorithms, we optimize *relative preferences* [41] instead of absolute objective values. We summarize our preference-driven MOCO in Appendix D and outline the complete pipeline in Fig. 2, which proceeds in three steps as follows.

**Generating preference pairs.** For each scalarized subproblem $(\mathcal{G}, \lambda_i)$ in the training batch, the policy $p_\theta$ samples two candidate solutions, $\pi^w$ and $\pi^l$. We denote $\pi^w \prec \pi^l$ if $\pi^w$ is preferred over $\pi^l$, as determined by the ordering of their scalarized objective values. This evaluator ranks the two solutions and designates the better one as the winning solution $\pi^w$ and the other as the losing solution $\pi^l$. A binary preference label $y$ is then assigned, where $y = 1$ if $\pi^w \prec \pi^l$, and $y = 0$ otherwise. This label serves as the supervision signal required for the preference-driven MOCO.

**Defining an implicit reward.** Distinct from DRL training paradigms that rely on raw objective values, POCCO treats the average log-likelihood of a solution as an implicit reward $f_\theta$, directly relating preferences between solutions to their policy probabilities. This reward is inherently normalized by sequence length $|\pi|$, thereby mitigating length bias between winning and losing solution pairs.

$$f_\theta(\pi|\mathcal{G}, \lambda_i) = \frac{1}{|\pi|} \log p_\theta(\pi|\mathcal{G}, \lambda_i) = \frac{1}{|\pi|} \sum_{t=1}^{|\pi|} \log p_\theta(\pi_t|\pi_{<t}, \mathcal{G}, \lambda_i). \tag{3}$$

**Learning from pairwise comparisons.** We formulate preference learning (PL) as a probabilistic binary classification problem through the BT model. Specifically, the BT model is a pairwise preference framework that uses a function $g_\theta(\cdot)$ to map reward differences into preference probabilities. It assigns each solution a strength proportional to its implicit reward (defined in Eq. (3)) and predicts the probability $g_\theta$ that the winning solution outranks the losing one:

$$g_\theta\big(\pi^w \prec \pi^\ell \mid \mathcal{G}, \lambda_i\big) = \sigma\big(\beta\,[\,f_\theta(\pi^w \mid \mathcal{G}, \lambda_i) - f_\theta(\pi^\ell \mid \mathcal{G}, \lambda_i)\,]\big), \tag{4}$$

where $\sigma(\cdot)$ is the sigmoid function, and $\beta > 0$ is a fixed temperature that controls the sharpness with which the model distinguishes between unequal rewards. We maximize the likelihood of the collected preferences, yielding the following loss function:

$$\mathcal{L}(\theta|p_\theta, \mathcal{G}, \lambda_i, \pi^w, \pi^l) = -y \log \sigma\big(\beta[\frac{\log p_\theta(\pi^w|\mathcal{G}, \lambda_i)}{|\pi^w|} - \frac{\log p_\theta(\pi^l|\mathcal{G}, \lambda_i)}{|\pi^l|}]\big). \tag{5}$$

In practice, we collect multiple $(\pi^w, \pi^\ell)$ pairs per update, sum their losses, and backpropagate through $p_\theta$. By maximizing the log-likelihood of $g_\theta(\pi^w \prec \pi^\ell \mid \mathcal{G}, \lambda_i)$, the model is encouraged to assign higher probabilities to preferred solutions $\pi^w$ over less preferred ones $\pi^l$.

## 5 Experiments

### 5.1 Experimental settings

**Training.** We conduct extensive experiments to evaluate the effectiveness of the proposed POCCO across various MOCOPs, including multi-objective traveling salesman problem (MOTSP) [39], multi-objective capacitated vehicle routing problem (MOCVRP) [64], and multi-objective knapsack problem (MOKP) [23]. In MOTSP, the goal is to find a tour that visits all nodes exactly once, while minimizing multiple total path lengths, each computed based on a distinct set of coordinates associated with a specific objective. Regarding MOCVRP, a fleet of vehicles with limited capacity must serve all customer nodes and return to the depot, with the objectives of minimizing the total travel distance and the length of the longest individual route. As for MOKP, the problem involves selecting a subset of items, each with a weight and multiple objective-specific values. The objective is to maximize all objective values simultaneously, while ensuring the total weight stays within the knapsack capacity. In this work, we consider three commonly used problem sizes: $n = 20/50/100$ for MOTSP and MOCVRP, and $n = 50/100/200$ for MOKP.

**Hyperparameters.** We implement POCCO on top of two SOTA neural MOCO methods, CNH [14] and WE-CA [6], resulting in POCCO-C and POCCO-W, respectively. Most hyperparameters are aligned with those used in the original CNH and WE-CA implementations. Both models are trained for 200 epochs, with each epoch processing 100,000 randomly sampled instances and a batch size of $B = 64$. We use the Adam optimizer [28] with a learning rate of $3 \times 10^{-4}$ and a weight decay of $10^{-6}$. The $N$ weight vectors used for decomposition are generated following [10], with $N = 101$ for $\kappa = 2$ and $N = 105$ for $\kappa = 3$.

**Baselines.** We compare POCCO with a broad range of baseline methods across three categories, all employing weighted-sum (WS) scalarization to ensure fair comparison: (1) Single-model neural MOCO approaches: This includes **PMOCO** [38], and recent SOTA methods **CNH** [14], and **WE-CA** [6]. Both CNH and WE-CA are unified models trained across problem size $n \in \{20, 21, \cdots, 100\}$ (except $n \in \{50, 51, \cdots, 200\}$ for Bi-KP) (2) Multi-model neural MOCO approaches: This category covers methods like **DRL-MOA** [34], **MDRL** [67], and **EMNH** [7]. Specifically, DRL-MOA trains a separate POMO model for each of the $N$ subproblems, with the first model trained for 200 epochs and the rest fine-tuned for 5 epochs each using parameter transfer. MDRL and EMNH both initialize from a shared pretrained meta-model and fine-tune $N$ subproblem-specific models using the same network structure and training settings as in [7]. (3) Non-learnable approaches, including classical MOEAs and other problem-specific heuristics: **MOEA/D** [65] and **NSGA-II** [12], each run for 4,000 iterations, serve as representative decomposition-based and dominance-based MOEAs, respectively. MOCOP-specific MOEAs such as **MOGLS** [24], configured with 4,000 iterations and 100 local search steps per iteration, and **PPLS/D-C** [49], run for 200 iterations, are also considered. These methods use 2-opt heuristics for MOTSP and MOCVRP, and a greedy transformation heuristic [23] for MOKP. Finally, **WS-LKH** and **WS-DP** combine weighted-sum scalarization with powerful solvers, with LKH [20, 51] used for MOTSP and dynamic programming applied to MOKP.

**Inference.** We evaluate all methods using three metrics: average hypervolume (HV) [54], average gap, and total runtime per instance set. HV is a widely used indicator in multi-objective optimization that reflects both the convergence and diversity of the solution set. A higher HV indicates better performance. To ensure consistency, HV values are normalized to the range $[0, 1]$ using the same reference point for all methods. The gap is defined as the relative difference between a method's HV and the HV of POCCO-W. Methods with the "-Aug" suffix apply instance augmentation [38] to further improve performance. To evaluate statistical significance, we use the Wilcoxon rank-sum test [55] at a 1% significance level. The best results and others that are not significantly worse are marked in bold, while the second-best and statistically similar results are underlined. All experiments are implemented in Python and conducted on a machine with NVIDIA Ampere A100-80GB GPUs and an AMD EPYC 7742 CPU. The code and dataset are publicly released for reproducibility.[2]

---

[2]https://github.com/mingfan321/POCCO

Table 1: Performance on BiTSP and MOCVRP Instances

| Method | Bi-TSP20 | | | Bi-TSP50 | | | Bi-TSP100 | | |
|---|---|---|---|---|---|---|---|---|---|
| | HV | Gap | Time | HV | Gap | Time | HV | Gap | Time |
| WS-LKH | 0.6270 | 0.00% | 10m | 0.6415 | 0.05% | 1.8h | **0.7090** | -0.17% | 6h |
| MOEA/D | 0.6241 | 0.46% | 1.7h | 0.6316 | 1.59% | 1.8h | 0.6899 | 2.53% | 2.2h |
| NSGA-II | 0.6258 | 0.19% | 6.0h | 0.6120 | 4.64% | 6.1h | 0.6692 | 5.45% | 6.9h |
| MOGLS | **0.6279** | -0.14% | 1.6h | 0.6330 | 1.37% | 3.7h | 0.6854 | 3.16% | 11h |
| PPLS/D-C | 0.6256 | 0.22% | 26m | 0.6282 | 2.12% | 2.8h | 0.6844 | 3.31% | 11h |
| DRL-MOA | 0.6257 | 0.21% | 6s | 0.6360 | 0.90% | 9s | 0.6970 | 1.53% | 16s |
| MDRL | 0.6271 | -0.02% | 5s | 0.6364 | 0.84% | 8s | 0.6969 | 1.54% | 14s |
| EMNH | 0.6271 | -0.02% | 5s | 0.6364 | 0.84% | 8s | 0.6969 | 1.54% | 15s |
| PMOCO | 0.6259 | 0.18% | 6s | 0.6351 | 1.04% | 12s | 0.6957 | 1.71% | 26s |
| CNH | 0.6270 | 0.00% | 13s | 0.6387 | 0.48% | 16s | 0.7019 | 0.83% | 33s |
| **POCCO-C** | 0.6275 | -0.08% | 14s | 0.6409 | 0.14% | 20s | 0.7047 | 0.44% | 42s |
| WE-CA | 0.6270 | 0.00% | 6s | 0.6392 | 0.41% | 9s | 0.7034 | 0.62% | 18s |
| **POCCO-W** | 0.6275 | -0.08% | 7s | 0.6411 | 0.11% | 14s | 0.7055 | 0.32% | 36s |
| MDRL-Aug | 0.6271 | -0.02% | 47s | 0.6408 | 0.16% | 1.8m | 0.7022 | 0.79% | 5.4m |
| EMNH-Aug | 0.6271 | -0.02% | 46s | 0.6408 | 0.16% | 1.8m | 0.7023 | 0.78% | 5.4m |
| PMOCO-Aug | 0.6270 | 0.00% | 39s | 0.6395 | 0.36% | 1.7m | 0.7016 | 0.88% | 5.8m |
| CNH-Aug | 0.6271 | -0.02% | 1.3m | 0.6410 | 0.12% | 3.9m | 0.7054 | 0.34% | 12m |
| **POCCO-C-Aug** | 0.6270 | 0.00% | 2.2m | 0.6416 | 0.03% | 4.0m | 0.7071 | 0.10% | 14m |
| WE-CA-Aug | 0.6271 | -0.02% | 1.3m | 0.6413 | 0.08% | 3.6m | 0.7066 | 0.17% | 12m |
| **POCCO-W-Aug** | 0.6270 | 0.00% | 2.2m | **0.6418** | 0.00% | 4.0m | 0.7078 | 0.00% | 14m |

| Method | MOCVRP20 | | | MOCVRP50 | | | MOCVRP100 | | |
|---|---|---|---|---|---|---|---|---|---|
| | HV | Gap | Time | HV | Gap | Time | HV | Gap | Time |
| MOEA/D | 0.4255 | 1.07% | 2.3h | 0.4000 | 2.63% | 2.9h | 0.3953 | 3.33% | 5.0h |
| NSGA-II | 0.4275 | 0.60% | 6.4h | 0.3896 | 5.16% | 8.8h | 0.3620 | 11.47% | 9.4h |
| MOGLS | 0.4278 | 0.53% | 9.0h | 0.3984 | 3.02% | 20h | 0.3875 | 5.23% | 72h |
| PPLS/D-C | 0.4287 | 0.33% | 1.6h | 0.4007 | 2.46% | 9.7h | 0.3946 | 3.50% | 38h |
| DRL-MOA | 0.4287 | 0.33% | 8s | 0.4076 | 0.78% | 12s | 0.4055 | 0.83% | 21s |
| MDRL | 0.4291 | 0.23% | 6s | 0.4082 | 0.63% | 13s | 0.4056 | 0.81% | 22s |
| EMNH | 0.4299 | 0.05% | 7s | 0.4098 | 0.24% | 12s | 0.4072 | 0.42% | 22s |
| PMOCO | 0.4267 | 0.79% | 6s | 0.4036 | 1.75% | 12s | 0.3913 | 4.30% | 22s |
| CNH | 0.4287 | 0.33% | 11s | 0.4087 | 0.51% | 15s | 0.4065 | 0.59% | 25s |
| **POCCO-C** | 0.4294 | 0.16% | 16s | 0.4101 | 0.17% | 25s | 0.4079 | 0.24% | 53s |
| WE-CA | 0.4290 | 0.26% | 7s | 0.4089 | 0.46% | 10s | 0.4068 | 0.51% | 21s |
| **POCCO-W** | 0.4294 | 0.16% | 8s | 0.4102 | 0.15% | 17s | 0.4084 | 0.12% | 46s |
| MDRL-Aug | 0.4294 | 0.16% | 12s | 0.4092 | 0.39% | 36s | 0.4072 | 0.42% | 2.8m |
| EMNH-Aug | **0.4302** | -0.02% | 12s | 0.4106 | 0.05% | 35s | 0.4079 | 0.24% | 2.8m |
| PMOCO-Aug | 0.4294 | 0.16% | 14s | 0.4080 | 0.68% | 42s | 0.3969 | 2.93% | 2.0m |
| CNH-Aug | 0.4299 | 0.05% | 21s | 0.4101 | 0.17% | 45s | 0.4077 | 0.29% | 1.9m |
| **POCCO-C-Aug** | **0.4302** | -0.02% | 31s | **0.4108** | 0.00% | 1.4m | 0.4086 | 0.07% | 2.4m |
| WE-CA-Aug | 0.4300 | 0.02% | 15s | 0.4103 | 0.12% | 36s | 0.4081 | 0.20% | 1.8m |
| **POCCO-W-Aug** | 0.4301 | 0.00% | 24s | **0.4108** | 0.00% | 1.2m | **0.4089** | 0.00% | 2.3m |

## 5.2 Experimental results

**Comparison analysis.** The comparison results are presented in Table 1 and Table 2. POCCO-W consistently achieves superior performance over WE-CA across all benchmark scenarios, establishing itself as the new SOTA results among neural MOCOP solvers. Similarly, POCCO-C outperforms CNH in every case. Both variants also surpass their augmentation-based counterparts, WE-CA-Aug and CNH-Aug, on Bi-TSP20 and Bi-CVRP100, highlighting POCCO's enhanced ability to explore the solution space and approximate high-quality Pareto fronts. When further combined with instance augmentation, POCCO demonstrates additional performance gains. Please note that POCCO with instance augmentation yields lower HV values compared with its non-augmented counterpart on Bi-TSP20. This is because decomposition-based methods focus on optimizing individual subproblems rather than ensuring overall solution diversity. While augmentation improves solution quality for specific subproblems, it may reduce the number of non-dominated solutions, resulting in a smaller HV. Compared with multi-model approaches that require training or fine-tuning separate models for each subproblem, POCCO delivers superior results while maintaining a single shared model. Notably, POCCO achieves better results on Bi-TSP50 than WS-LKH, a setting where previous neural solvers have consistently failed. In terms of efficiency, POCCO significantly reduces computational time.

Table 2: Performance comparison on Bi-KP and Tri-TSP Instances

| Method | Bi-KP50 | | | Bi-KP100 | | | Bi-KP200 | | |
|---|---|---|---|---|---|---|---|---|---|
| | HV | Gap | Time | HV | Gap | Time | HV | Gap | Time |
| WS-DP | 0.3561 | 0.03% | 22m | 0.4532 | 0.04% | 2h | 0.3601 | 0.06% | 5.8h |
| MOEA/D | 0.3540 | 0.62% | 1.6h | 0.4508 | 0.57% | 1.7h | 0.3581 | 0.61% | 1.8h |
| NSGA-II | 0.3547 | 0.42% | 7.8h | 0.4520 | 0.31% | 8.0h | 0.3590 | 0.36% | 8.4h |
| MOGLS | 0.3540 | 0.62% | 5.8h | 0.4510 | 0.53% | 10h | 0.3582 | 0.58% | 18h |
| PPLS/D-C | 0.3528 | 0.95% | 18m | 0.4480 | 1.19% | 47m | 0.3541 | 1.72% | 1.5h |
| DRL-MOA | 0.3559 | 0.08% | 8s | 0.4531 | 0.07% | 15s | 0.3601 | 0.06% | 32s |
| MDRL | 0.3530 | 0.90% | 7s | 0.4532 | 0.04% | 18s | 0.3601 | 0.06% | 35s |
| EMNH | 0.3561 | 0.03% | 7s | **0.4535** | -0.02% | 17s | **0.3603** | 0.00% | 48s |
| PMOCO | 0.3552 | 0.28% | 8s | 0.4523 | 0.24% | 22s | 0.3595 | 0.22% | 50s |
| CNH | 0.3556 | 0.17% | 16s | 0.4527 | 0.15% | 23s | 0.3598 | 0.14% | 55s |
| **POCCO-C** | 0.3560 | 0.06% | 20s | **0.4535** | -0.02% | 36s | **0.3603** | 0.00% | 1.4m |
| WE-CA | 0.3558 | 0.11% | 8s | 0.4531 | 0.07% | 16s | 0.3602 | 0.03% | 50s |
| **POCCO-W** | **0.3562** | 0.00% | 11s | 0.4534 | 0.00% | 26s | **0.3603** | 0.00% | 1.3m |

| Method | Tri-TSP20 | | | Tri-TSP50 | | | Tri-TSP100 | | |
|---|---|---|---|---|---|---|---|---|---|
| | HV | Gap | Time | HV | Gap | Time | HV | Gap | Time |
| WS-LKH | **0.4712** | 0.00% | 12m | **0.4440** | -0.07% | 1.9h | **0.5076** | -0.55% | 6.6h |
| MOEA/D | 0.4702 | 0.21% | 1.9h | 0.4314 | 2.77% | 2.2h | 0.4511 | 10.64% | 2.4h |
| NSGA-II | 0.4238 | 10.06% | 7.1h | 0.2858 | 35.59% | 7.5h | 0.2824 | 44.06% | 9.0h |
| MOGLS | 0.4701 | 0.23% | 1.5h | 0.4211 | 5.09% | 4.1h | 0.4254 | 15.73% | 13h |
| PPLS/D-C | 0.4698 | 0.30% | 1.4h | 0.4174 | 5.93% | 3.9h | 0.4376 | 13.31% | 14h |
| DRL-MOA | 0.4699 | 0.28% | 6s | 0.4303 | 3.02% | 9s | 0.4806 | 4.79% | 18s |
| MDRL | 0.4699 | 0.28% | 5s | 0.4317 | 2.70% | 10s | 0.4852 | 3.88% | 17s |
| EMNH | 0.4699 | 0.28% | 5s | 0.4324 | 2.55% | 10s | 0.4866 | 3.61% | 17s |
| PMOCO | 0.4693 | 0.40% | 5s | 0.4315 | 2.75% | 12s | 0.4858 | 3.76% | 33s |
| CNH | 0.4698 | 0.30% | 10s | 0.4358 | 1.78% | 14s | 0.4931 | 2.32% | 25s |
| **POCCO-C** | 0.4704 | 0.17% | 18s | 0.4393 | 0.99% | 17s | 0.4985 | 1.25% | 28s |
| WE-CA | 0.4707 | 0.11% | 5s | 0.4389 | 1.08% | 8s | 0.4975 | 1.45% | 17s |
| **POCCO-W** | 0.4710 | 0.04% | 6s | 0.4397 | 0.90% | 13s | 0.4985 | 1.25% | 23s |
| MDRL-Aug | **0.4712** | 0.00% | 4.2m | 0.4408 | 0.65% | 25m | 0.4958 | 1.78% | 1.6h |
| EMNH-Aug | **0.4712** | 0.00% | 4.2m | 0.4418 | 0.43% | 25m | 0.4973 | 1.49% | 1.6h |
| PMOCO-Aug | **0.4712** | 0.00% | 4.9m | 0.4409 | 0.63% | 28m | 0.4956 | 1.82% | 1.6h |
| CNH-Aug | 0.4704 | 0.17% | 8.5m | 0.4409 | 0.63% | 28m | 0.4996 | 1.03% | 1.6h |
| **POCCO-C-Aug** | 0.4706 | 0.13% | 8.9m | 0.4419 | 0.41% | 34m | 0.5023 | 0.50% | 2h |
| WE-CA-Aug | **0.4712** | 0.00% | 8.2m | 0.4432 | 0.11% | 29m | 0.5035 | 0.26% | 1.7h |
| **POCCO-W-Aug** | **0.4712** | 0.00% | 8.9m | 0.4437 | 0.00% | 33m | 0.5048 | 0.00% | 2h |

For example, POCCO-W-Aug solves Bi-TSP100 in only 14 minutes, while WE-LKH requires about 6.0 hours, with POCCO delivering comparable solution quality.

**Out-of-distribution size generalization analysis.** We evaluate the generalization ability of the models on out-of-distribution, benchmark instances KroAB100/150/200 [40]. All neural methods are trained on Bi-TSP100, except for CNH, WE-CA, and POCCO, which are trained across varying sizes $n \in \{20, 21, \ldots, 100\}$. The results, visualized in Fig. 3, show that POCCO-W-Aug consistently achieves the best generalization performance compared with other neural baselines. POCCO-C-Aug also outperforms CNH-Aug across all evaluated scenarios. Full benchmark results and experiments on larger-scale instances Bi-TSP150/200 are provided in Appendix I and Appendix J, respectively.

**Effectiveness of PL.** We assess the training efficiency of PL by comparing it to REINFORCE on the WE-CA and POCCO-W models using the Bi-TSP100 dataset. As shown by the validation curves in Fig. 4a, PL achieves faster convergence despite identical network architectures. Notably, for WE-CA, training with PL for 100 epochs reaches performance comparable to 200 epochs of REINFORCE. Similar improvements are observed for POCCO-W. These results demonstrate that PL accelerates the training process and achieves better performance with fewer training epochs.

**Effectiveness of the CCO block.** To evaluate the impact of the CCO block's structure and placement, we compare POCCO-W with WE-CA and several POCCO variants: POCCO-E (CCO inserted in the encoder replacing the FF layer), POCCO-D (CCO replacing the final linear layer of MHA in the decoder, using MLP experts), POCCO-ME (replacing CCO with a standard MoE layer), POCCO-MD (replacing CCO with three MoD layers), POCCO-MED (using MoE in the encoder and MoD in place of CCO in the decoder), and POCCO-Emp (replacing the identity expert in CCO

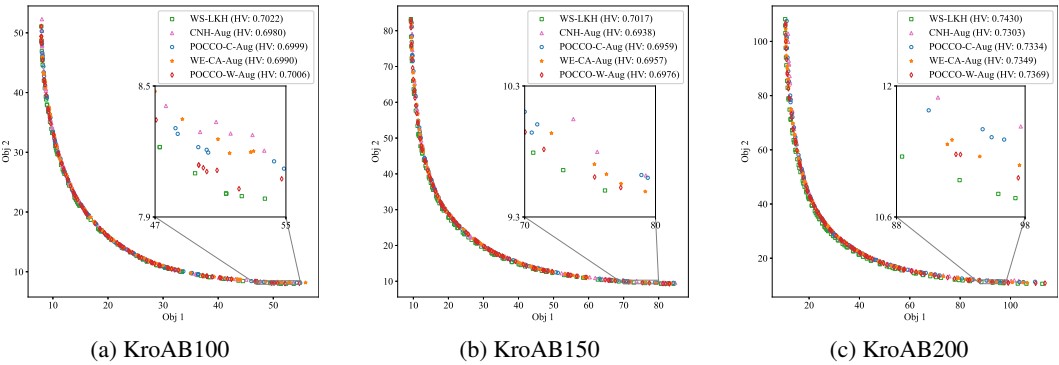

(a) KroAB100      (b) KroAB150      (c) KroAB200

Figure 3: Pareto fronts of benchmark instances.

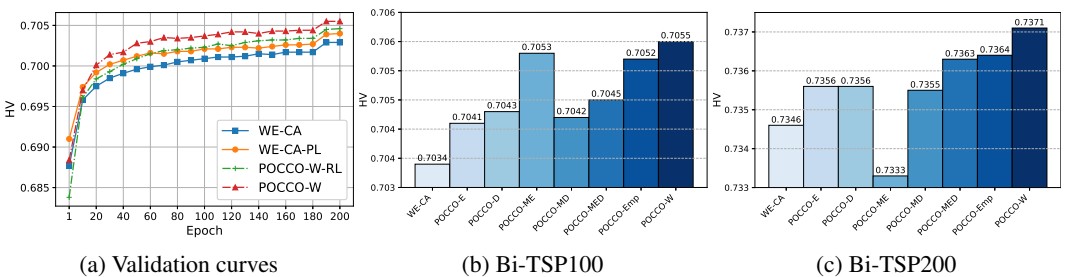

(a) Validation curves      (b) Bi-TSP100      (c) Bi-TSP200

Figure 4: Ablation study:(a) validates the effectiveness of PL; (b) and (c) verify the effects of different CCO block variants.

with an empty expert). As shown in Fig. 4b, all variants outperform WE-CA on the in-distribution Bi-TSP100, with POCCO-W, POCCO-ME, and POCCO-Emp achieving the most notable gains. On the out-of-distribution Bi-TSP200 in Fig. 4c, only POCCO-W and POCCO-Emp maintain strong performance, while POCCO-ME performs worst, even underperforming WE-CA. These results highlight the importance of both the structure and placement of the CCO block for achieving strong generalization across in- and out-of-distribution settings.

**Hyperparameter study.** We conduct experiments to examine the impact of different key hyperparameters on POCCO's performance. As detailed in Appendix L, the number of CCO block layers, the number of experts, the $\text{Top } k$ value, and the temperature parameter $\beta$ all influence model effectiveness. For the problems studied, the desirable settings, as identified based on empirical results, are: one CCO block layer, four FF experts and one ID expert, $\text{Top } k = 2$, $\beta = 3.5$ for bi-objective tasks, and $\beta = 4.5$ for tri-objective tasks.

## 6 Conclusion

This paper presents POCCO, a plug-and-play framework tailored for MOCOPs, which adaptively routes subproblems through different model structures and leverages PL for more effective training. POCCO is integrated into two SOTA neural solvers, and extensive experiments demonstrate its effectiveness. Ablation studies further highlight the necessity of both CCO block and PL, and reveal the critical impact of the design and placement of the CCO block. We acknowledge certain limitations, such as the limited capability to address real-world MOCOPs with complex constraints or large problem sizes. Addressing these challenges may require constraint-handling mechanisms [5] or divide-and-conquer [36] strategies, which we leave for future work.

## Acknowledgments and Disclosure of Funding

We thank the anonymous reviewers and (S)ACs of NeurIPS 2025 for their constructive comments and dedicated service to the community.

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

# Appendix

## A Details of MOCOP

Here we elaborate on the problem definitions for the three typical MOCOPs, i.e., MOTSP, MOCVRP, and MOKP.

**MOTSP.** An MOTSP instance involves multiple cost matrices, aiming to identify a set of tours, i.e., node sequences, that are Pareto optimal. For instance, a $\kappa$-objective TSP instance $\mathcal{G}$ with $n+1$ nodes is featured by cost matrices $C^i = (c^i_{j,k})$, with $i \in \{1, \cdots, \kappa\}$ and $j, k \in \{0, \cdots, n\}$. The $\kappa$ objectives are defined as follows,

$$\min_{\pi \in \mathcal{X}} F(\pi) = \min(f_1(\pi), f_2(\pi), \cdots, f_\kappa(\pi)),$$

$$\text{with } f_i(\pi) = c^i_{\pi_n \pi_0} + \sum_{j=0}^{n-1} c^i_{\pi_j \pi_{j+1}}, \tag{6}$$

where $\pi = (\pi_0, \pi_2, \cdots, \pi_n)$ with $\pi_j \in \{0, \cdots, n\}$. $\mathcal{X}$ represents all feasible solutions (i.e., tours), ensuring each node is visited exactly once. This paper considers the Euclidean MOTSP following [16, 34, 38]. Each node $j$ has a $2\kappa$-dim feature vector $o_j = [loc^1_j, loc^2_j, \cdots, loc^\kappa_j]$, where $loc^i_j \in \mathbb{R}^2$ is the coordinate for the $i$-th objective. The objective $f_i(\pi)$ is defined as $f_i(\pi) = \|loc^i_{\pi_n} - loc^i_{\pi_0}\|_2 + \sum_{j=0}^{n-1} \|loc^i_{\pi_j} - loc^i_{\pi_{j+1}}\|_2$.

**MOCVRP.** An MOCVRP instance consists of $n$ customer nodes and one depot node. Each node has a 3-dim feature vector $o_j = [loc_j, \delta_j]$, where $loc_j$ and $\delta_j$, for $j \in \{0, \cdots, n\}$, correspond to the coordinates and demand of node $j$. Notably, for the depot node, $o_0 = [loc_0, \delta_0]$, the demand $\delta_0$ is set to 0. Vehicles with a capacity of $\mathcal{Q}$ ($\mathcal{Q} > \delta_i$) are employed to serve all customers in multiple routes, with each route commencing and concluding at the depot. The problem must satisfy the following constraints: 1) Each customer is visited exactly once, and 2) The total demand of customers in each route must not exceed the vehicle's capacity. In our study, we focus on the bi-objective CVRP, aligning with prior research [31, 38]. Specifically, we aim to minimize two objectives: the total length of all routes and the length of the longest route (i.e., the makespan).

**MOKP.** An MOKP instance consists of $n+1$ items, where each item $j$ is characterized by a 2-dim feature vector $o_j = [w_j, p_j]$, representing its weight $w_j$ and profit vector $p_j$, for $j \in \{0, \cdots, n\}$. The profit vector $p_j \in \mathbb{R}^\kappa$ corresponds to $\kappa$ distinct objective values associated with item $j$. A knapsack with a capacity of $\mathcal{C}$ is provided, where $\mathcal{C} > w_j$ for each item. The goal is to select a subset of items to place into the knapsack while satisfying the following constraint: the total weight of selected items must not exceed the knapsack capacity $\mathcal{C}$. In our study, we focus on the bi-objective knapsack problem, consistent with previous research [40, 68]. Specifically, we aim to simultaneously maximize two objectives: the sum of the first and second profit components across the selected items.

## B Theoretical Analysis

We derive the loss function and gradient for our PL objective and contrast it with the REINFORCE gradient. Our theoretical (and empirical) analyses demonstrate that PL yields significantly lower gradient variance, which contributes to faster and more stable convergence.

### B.1 Formulation of Loss Function

Different from the RL, our PL method exploits the preference relations between generated solutions according to their objective. The explicit preference $f^*(\pi|\mathcal{G}, \lambda)$ is defined as the negation of the scalarized objective. We contruct a preference pair, denoted as $(\pi^w, \pi^\ell)$, along with a binary preference label $y$, where $y = 1$ if $\pi^w \prec \pi^l$ (i.e., $f^*(\pi^w|\mathcal{G}, \lambda) > f^*(\pi^\ell|\mathcal{G}, \lambda)$), and $y = 0$ otherwise.

For the policy $p_\theta(\pi|\mathcal{G}, \lambda)$ used to construct solution $\pi$, its implicit preference is defined as the average log-likelihood: $f_\theta(\pi \mid \mathcal{G}, \lambda) = \frac{1}{|\pi|} \log p_\theta(\pi \mid \mathcal{G}, \lambda)$. For a preference pair $(\pi^w, \pi^\ell)$, the preference distribution is modeled using the Bradley-Terry (BT) ranking objective and implicit preferences:

$$g_\theta(\pi^w \prec \pi^\ell \mid \mathcal{G}, \lambda) = \sigma(\beta[f_\theta(\pi^w \mid \mathcal{G}, \lambda) - f_\theta(\pi^\ell \mid \mathcal{G}, \lambda)]),$$

where $\sigma(\cdot)$ is the sigmoid function, and $\beta > 0$ is a fixed temperature that controls the sharpness with which the model distinguishes between unequal rewards. By maximizing the log-likelihood of $g_\theta(\pi^w \prec \pi^\ell \mid \mathcal{G}, \lambda)$, the model is encouraged to assign higher probabilities to preferred solutions $\pi^w$ compared with less preferred solutions $y^\ell$. We can derive the PL loss function:

$$\mathcal{L}_{PL}(\theta|p_\theta, \mathcal{G}, \lambda, \pi^w, \pi^l) = -y \log \sigma(\beta[\frac{\log p_\theta(\pi^w|\mathcal{G}, \lambda)}{|\pi^w|} - \frac{\log p_\theta(\pi^l|\mathcal{G}, \lambda)}{|\pi^l|}]),$$

PL directly distinguishes optimization signals based on exact preferences, and the strength of the optimization signal correlates with the difference in log-likelihood, requiring the model to maximize the probability gap between $\pi^w$ and $\pi^\ell$.

## B.2 Gradient Analysis

Let $z$ denote the argument of the sigmoid function:
$$z = \beta[\frac{\log p_\theta(\pi^w|\mathcal{G}, \lambda)}{|\pi^w|} - \frac{\log p_\theta(\pi^l|\mathcal{G}, \lambda)}{|\pi^l|}].$$

The gradient of $\mathcal{L}_{PL}$ with respect to $\theta$ is:
$$\nabla_\theta \mathcal{L}_{PL} = \frac{\partial \mathcal{L}_{PL}}{\partial z} \cdot \nabla_\theta z.$$

Derivative of $-y \log \sigma(z)$ becomes:
$$\frac{\partial \mathcal{L}_{PL}}{\partial z} = -y(1 - \sigma(z)).$$

Gradient of $z$ with respect to $\theta$:
$$\nabla_\theta z = \beta[\frac{1}{|\pi^w|}\nabla_\theta \log p_\theta(\pi^w|\mathcal{G}, \lambda) - \frac{1}{|\pi^\ell|}\nabla_\theta \log p_\theta(\pi^\ell|\mathcal{G}, \lambda)].$$

Combining these, the total gradient becomes:
$$\nabla_\theta \mathcal{L}_{PL} = -y\beta(1 - \sigma(z))[\frac{1}{|\pi^w|}\nabla_\theta \log p_\theta(\pi^w|\mathcal{G}, \lambda) - \frac{1}{|\pi^\ell|}\nabla_\theta \log p_\theta(\pi^\ell|\mathcal{G}, \lambda)].$$

The gradient increases the likelihood of $\pi^w$ and decreases the likelihood of $\pi^\ell$. For comparison, we analyze the loss function in the REINFORCE algorithm here:
$$\mathcal{L}_{RL}(\pi|\mathcal{G}, \lambda) = -(\mathcal{R}(\pi) - b)\log p_\theta(\pi|\mathcal{G}, \lambda),$$

where $b$ is a baseline to distinguish positive or negative optimization signals for each solution $\pi$. The gradient of the REINFORCE algorithm is:

$$\nabla_\theta \mathcal{L}_{RL} = -(\mathcal{R}(\pi) - b)\nabla_\theta \log p_\theta(\pi|\mathcal{G}, \lambda).$$

REINFORCE relies on *absolute* returns $\mathcal{R}(\pi)$, whose large variance propagates directly to the gradient. In contrast, $\mathcal{L}_{PL}$ uses *relative* returns inside a sigmoid, yielding the difference of two normalized log-likelihoods. This pairwise, centered signal reduces gradient variance and produces smoother, more stable updates, leading to faster and more reliable convergence. This theoretical insight is further supported by our empirical analyses. We provide concrete evidence of this in Table 3, which reports the average gradient variance in the first five batches of training POCCO-W under both RL and PL frameworks. Across three MOTSP settings, PL consistently exhibits two to four orders of magnitude lower variance, leading to faster convergence and more stable optimization dynamics.

Table 3: Gradient variance in REINFORCE vs. preference learning.

| Problem Size | Batch | RL Variance | PL Variance |
|---|---|---|---|
| MOTSP20 | 1 | 0.054648 | 0.000314 |
| | 2 | 0.039951 | 0.000252 |
| | 3 | 0.019038 | 0.000191 |
| | 4 | 0.009093 | 0.000115 |
| | 5 | 0.010742 | 0.000104 |
| MOTSP50 | 1 | 0.474784 | 0.000142 |
| | 2 | 0.220530 | 0.000077 |
| | 3 | 0.140275 | 0.000048 |
| | 4 | 0.092114 | 0.000040 |
| | 5 | 0.065286 | 0.000031 |
| MOTSP100 | 1 | 10.124832 | 0.000059 |
| | 2 | 6.355499 | 0.000039 |
| | 3 | 3.461700 | 0.000032 |
| | 4 | 1.755804 | 0.000021 |
| | 5 | 1.111246 | 0.000014 |

## C  MOCOP Solvers with POCCO

Existing neural MOCOP solvers are based on a transformer-based architecture that consists of an encoder and a decoder. The encoder is used to generate embeddings for all nodes based on the instance $\mathcal{G}$ and the weight vector $\lambda$. The decoder is used to decode a sequence of actions based on these embeddings in an iterative fashion. To demonstrate the versatility of our POCCO, we integrate it into two SOTA neural methods, WE-CA [6] and CNH [14], yielding POCCO-W and POCCO-C.

### C.1  POCCO-W

Given an instance $\mathcal{G}$ comprising $n + 1$ nodes with $Z$-dimensional features $\{o_i\}_{i=0}^n \subset \mathbb{R}^Z$ and a weight vector $\lambda$, first obtains initial embeddings by applying separate linear projections to the node features and the weight vector:

$$h_i^0 = W^o o_i + b^o, \quad \forall i \in \{0, \ldots, n\}, \quad h_\lambda^0 = W^\lambda \lambda + b^\lambda, \tag{7}$$

where $W^o \in \mathbb{R}^{d \times Z}$, $W^\lambda \in \mathbb{R}^{d \times \kappa}$, and $b^o, b^\lambda \in \mathbb{R}^d$ are learnable parameters, with embedding dimension $d = 128$. POCCO-W integrates the weight embedding into each node embedding in a feature-wise fashion within the encoder. To ensure harmonious interaction, the weight and node embeddings are updated simultaneously. The encoder itself comprises $L = 6$ transformer layers, each layer applying a conditional attention sublayer, followed by a residual Add & Norm (skip connection with instance normalization), a fully connected feed-forward sublayer, and a second Add & Norm.

Specifically, the conditional attention sublayer first conditions node embeddings on the weight embedding via a feature-wise affine transform:

$$\gamma = W^\gamma h_\lambda^{l-1}, \quad \beta = W^\beta h_\lambda^{l-1}, \quad h_i' = \gamma \circ h_i^{l-1} + \beta, \quad \forall i \in \{0, \ldots, n\}, \tag{8}$$

where $W^\gamma$ and $W^\beta$ are trainable matrices; $\circ$ denotes element-wise multiplication. Then, the weight and node embeddings are updated via the Multi-Head Attention (MHA) mechanism with 8 heads and an Add & Norm, as follows:

$$\hat{h}_\lambda = \text{IN}\big(h_\lambda^{l-1} + \text{MHA}\big(h_\lambda^{l-1}, \{h_\lambda^{l-1}, h_0', \ldots, h_n'\}\big)\big), \tag{9}$$

$$\hat{h}_i = \text{IN}\big(h_i^{l-1} + \text{MHA}\big(h_i', \{h_\lambda^{l-1}, h_0', \ldots, h_n'\}\big)\big), \quad \forall i \in \{0, \ldots, n\}. \tag{10}$$

Afterwards, a fully connected feed-forward sublayer and another Add & Norm are employed to yield the weight embedding $h_\lambda^l$ and the node embeddings $\{h_i^l\}_{i=0}^n$, as follows:

$$h_i^l = \text{IN}\big(\hat{h}_i + \text{FF}(\hat{h}_i)\big), \tag{11}$$

$$h_\lambda^l = \text{IN}\big(\hat{h}_\lambda + \text{FF}(\hat{h}_\lambda)\big), \tag{12}$$

Given the eventual node embeddings $\{h_i\}_{i=0}^n$ and weight embedding $h_\lambda$ output by the encoder, the decoder autoregressively computes the probability of node selection over $T$ steps. At decoding step $t \in \{1, \ldots, T\}$, the advanced context vector $h_c$ is produced by an MHA layer with 8 heads based on the context embedding $v_c$ and eventual node embeddings as follows:

$$h_c = \text{MHA}\Big(v_c, \{h_\lambda, h_0, \ldots, h_n\}\Big), \tag{13}$$

**Context embedding.** For MOTSP, the context embedding $v_c$ is obtained by concatenating the embeddings of the first and last visited nodes, and all previously visited nodes are masked when computing selection probabilities. In MOCVRP, $v_c$ consists of the embedding of the last visited node together with the remaining vehicle capacity, while nodes that have already been visited or whose demand exceeds the remaining capacity are masked. In MOKP, the context embedding combines the graph embedding $\bar{h} = \frac{1}{n+1}\sum_{i=0}^n h_i$ with the remaining knapsack capacity, masking both items that are already selected and those whose weight exceeds the remaining capacity when computing selection probabilities.

The context vector $h_c$ is fed through the CCO block to produce a glimpse $g_c$ based on Eq. 2 (i.e., $g_c = \text{CCO}(h_c)$). This glimpse $g_c$ is then used in a compatibility layer to compute unnormalized compatibility scores $\alpha$ as follows:

$$\alpha_i = \begin{cases} -\infty, & \text{if node } i \text{ is masked}, \\ C \cdot \tanh\left(\frac{g_c^\top (W^K h_i)}{\sqrt{d}}\right), & \text{otherwise}, \end{cases} \tag{14}$$

where $C$ is set to 50 and $W^K$ is a learnable weight matrix. Finally, the node-selection probability for the scalarized subproblem is given by:

$$P_\theta(\pi_t \mid \pi_{1:t-1}, s) = \text{Softmax}(\alpha). \tag{15}$$

## C.2 POCCO-C

POCCO-C mirrors the overall design of POCCO-W but (i) replaces every conditional-attention sublayer in the encoder with a dual-attention sublayer and (ii) enriches the context vector $h_c$ via a problem-size embedding (PSE) layer. Specifically, each of the $L$ encoder layers in POCCO-C consists of a dual-attention sublayer, followed by a residual Add & Norm, a fully connected feed-forward sublayer, and a second Add & Norm. Concretely, at layer $l$ the dual-attention and first Add & Norm operate as:

$$\hat{h}_\lambda = \text{IN}\big(h_\lambda^{l-1} + \text{MHA}\big(h_\lambda^{l-1}, \{h_\lambda^{l-1}, h_0^{l-1}, \ldots, h_n^{l-1}\}\big)\big), \tag{16}$$

$$\hat{h}_i' = \text{MHA}\big(h_i^{l-1}, \{h_\lambda^{l-1}, h_0^{l-1}, \ldots, h_n^{l-1}\}\big) + \text{MHA}\big(h_i^{l-1}, \{h_\lambda^{l-1}\}\big), \quad \forall i \in \{0, \ldots, n\}. \tag{17}$$

$$\hat{h}_i = \text{IN}\big(h_i^{l-1} + \hat{h}_i'\big), \quad \forall i \in \{0, \ldots, n\}. \tag{18}$$

Besides, POCCO-C employs the sinusoidal encoding based on sine and cosine functions to yield the PSE as follows,

$$\begin{aligned} \text{PSE}(\xi, 2i) &= sin(\xi/10000^{2i/d}), \\ \text{PSE}(\xi, 2i+1) &= cos(\xi/10000^{2i/d}), \end{aligned} \tag{19}$$

where $\xi(\xi = n + 1)$ and $i$ ($i \in \{0, \cdots, 63\}$) mean the problem size and dimension. The resulting d-dimensional PSEs are then processed using two linear layers with trainable matrices $W_{\xi 1} \in \mathbb{R}^{d \times 2d}$ and $W_{\xi 2} \in \mathbb{R}^{2d \times d}$. POCCO-C injects the size information by adding the results from PSE to $\{h_i\}_{i=0}^n$ from the encoder, such that,

$$h_i^\xi = h_i + h_\xi, \quad \text{with } h_\xi = (\text{PSE}(\xi, \cdot)W_{\xi 1})W_{\xi 2}. \tag{20}$$

Then, POCCO-C produces the advanced context vector $h_c$ based on the context embedding $v_c$ and the size-injected node embeddings $\{h_i^\xi\}_{i=0}^n$ through a MHA layer with 8 heads as follows,

$$h_c = \text{MHA}\Big(v_c, \{h_0^\xi, \ldots, h_n^\xi\}\Big). \tag{21}$$

**Algorithm 1** Preference-driven MOCO

---

**Input**: Instance distribution $\tilde{\mathcal{G}}$, weight vector distribution $\tilde{\lambda}$, number of training steps $E$, batch size $B$, number of tours $K$ per subproblem;
**Output**: The trained policy network $\theta$;

1: Initialize policy network $\theta$.
2: **for** $e = 1$ to $E$ **do**
3:     $\lambda_b \sim \textsc{SampleWeightVector}(\tilde{\lambda}); \mathcal{G}_b \sim \textsc{SampleInstance}(\tilde{\mathcal{G}}), \quad \forall b \in \{1, \cdots, B\}$
4:     $\pi^{j,b} \sim \textsc{SampleSolutions}(p_\theta(\cdot|\mathcal{G}_b, \lambda_b)), \forall j \in \{1, \cdots, K\}, \quad \forall b \in \{1, \cdots, B\}$
5:     $y_{j,p}^b \sim \textsc{PairwisePreference}(1_{[\pi^{j,b} \prec \pi^{p,b}]}), \quad \forall j, p \in \{1, \cdots K\}, \forall b \in \{1, \cdots, B\}$
6:     Calculate gradient $\nabla_\theta \mathcal{L}(\theta)$ according to Eq. (5)
7:     $\theta \leftarrow \text{ADAM}(\theta, \nabla_\theta \mathcal{L}(\theta))$
8: **end for**

---

### C.3 Gating Mechanism

Our CCO block employs a subproblem-level gating mechanism to dynamically route each context vector $h_c$ to a subset of experts from a pool consisting of 4 FF experts and 1 ID expert (thus $m = 4$). Let $d$ be the hidden dimension and $W_G \in \mathbb{R}^{d \times (m+1)}$ denote the trainable gating weight matrix. Given a batch of context vectors $X = \{h_c^b\}_{b=1}^B \in \mathbb{R}^{B \times d}$, where $B$ is the batch size, the gating network computes a score matrix: $H = X \cdot W_G \in \mathbb{R}^{B \times (m+1)}$, where each element $H_b^j$ represents the affinity or preference (score) of subproblem $b$ towards expert $j$.

For each subproblem, the router selects the Top-$k$ highest-scoring experts from the score vector $H_b^j$. These selected experts are then activated, and their outputs are weighted by a softmax-normalized version of their scores. In our POCCO with $k = 2$, each subproblem is routed to its two highest-scoring experts.

This Top-$k$ routing plays a critical role by reducing computation, as only $k$ experts are activated per subproblem (i.e., sparse activation widely used in MoE structure [69]). It also encourages expert specialization, since different subproblems (defined by context and weight vectors) tend to activate different subsets of experts. Additionally, it allows the inclusion of a parameter-free ID expert, which is often selected when minimal transformation is needed. This enables the router to learn when it is appropriate to leave a representation unchanged.

## D Training Algorithm

The training algorithm is provided in Algorithm 1. To train the model with preference learning, we first sample a batch of instances and weight vectors $\{(\mathcal{G}_b, \lambda_b)\}_{b=1}^B$ (as in Line 3). Then, for each scalarized subproblem $(\mathcal{G}_b, \lambda_b)$, we construct a set of winning–losing solution pairs $(\pi^{j,b}, \pi^{p,b}), \forall j, p \in \{1, \cdots, K\}$ (as in Lines 4-5). Training then proceeds by maximizing the likelihood of the winning solutions while minimizing that of the losing ones (as in Lines 6-7).

## E Decomposition Approaches

The major decomposition techniques include weighted-sum, Tchebycheff, and penalty-based boundary intersection (PBI) approaches [42, 65], respectively.

**Weighted-sum Approach.** Given an MOCOP, the $i$th subproblem (i.e., SOCOP) is defined with the $i$th weight vector $\lambda_i$, such that,

$$\min \quad g_w(\pi|\lambda_i) = \sum_{j=1}^\kappa \lambda_i^j f_j(\pi), \ \pi \in \mathcal{X} \tag{22}$$

**Tchebycheff Approach.** It minimizes the maximal distance between objectives and the ideal reference point, such that,

$$\min \quad g_t(\pi|\lambda_i, z^*) = \max_{1 \le j \le \kappa} \left\{ \lambda_i^j |f_j(\pi) - z_j^*| \right\}, \ \pi \in \mathcal{X} \tag{23}$$

where $z^* = (z_1^*, \cdots, z_\kappa^*)^\top$ signifies the ideal reference point with $z_j^* \leq \min\{f_j(\pi)|\pi \in \mathcal{X}\}$. It guarantees that the optimal solution in Eq. (23) under a specific (but unknown) weight vector $\lambda_i$ could be a Pareto optimal solution [9].

**PBI Approach.** This approach formulates the $i$th subproblem of an MOCOP as follows,

$$
\begin{aligned}
\min \quad & g_p(\pi|\lambda) = d_1 + \alpha d_2 \\
\text{where} \quad & d_1 = \frac{\left\|(F(\pi)-z^*)^\top \lambda\right\|}{\|\lambda\|} \\
& d_2 = \|F(\pi) - (z^* + d_1\lambda)\|, \pi \in \mathcal{X}
\end{aligned}
\tag{24}
$$

where $\alpha > 0$ is a preset penalty item and $z^*$ is the ideal reference point as defined in the Tchebycheff approach.

## F  Hypervolume

Hypervolume (HV) is a widely used indicator to evaluate approximate Pareto solutions to MOCOPs. Formally, the HV for a set of solutions $\mathcal{P}$ is defined as the volume of the subspace, which is weakly dominated by the solutions in $\mathcal{P}$ and bounded by a reference point $r^*$, such that,

$$
\text{HV}(\mathcal{P}) = \zeta^\kappa(\{r \in \mathbb{R}^\kappa | \exists\, \pi \in \mathcal{P}, \pi \prec r \prec r^*\}),
\tag{25}
$$

where $\zeta^\kappa$ denotes the Lebesgue measure on the $\kappa$-dimensional space, i.e., the volume for a $m$-dimensional subspace [54]. Since the range of objective values varies among different problems, we report the normalized HV $\bar{H}(\mathcal{P}) = \text{HV}(\mathcal{P})/\prod_{i=1}^\kappa |r_i^* - z_i|$, where the ideal point $z = (z_1, \ldots, z_\kappa)$ satisfies $z_i < \min\{f_i(\pi) \mid \pi \in \mathcal{P}\}$ (or $z_i > \max\{f_i(\pi) \mid \pi \in \mathcal{P}\}$ for maximization), $\forall\, i \in \{1, \ldots, \kappa\}$. All methods share the same $r^*$ and $z$ for an MOCOP, as given in Table 4, and we report the average $\bar{H}(\mathcal{P})$ over all test instances in this paper.

Table 4: Reference points and ideal points.

| Problem | Size | $r^*$ | $z$ |
|---|---|---|---|
| Bi-TSP | 20 | $(20, 20)$ | $(0, 0)$ |
| | 50 | $(35, 35)$ | $(0, 0)$ |
| | 100 | $(65, 65)$ | $(0, 0)$ |
| | 150 | $(85, 85)$ | $(0, 0)$ |
| | 200 | $(115, 115)$ | $(0, 0)$ |
| Bi-CVRP | 20 | $(30, 4)$ | $(0, 0)$ |
| | 50 | $(45, 4)$ | $(0, 0)$ |
| | 100 | $(80, 4)$ | $(0, 0)$ |
| Bi-KP | 50 | $(5, 5)$ | $(30, 30)$ |
| | 100 | $(20, 20)$ | $(50, 50)$ |
| | 200 | $(30, 30)$ | $(75, 75)$ |
| Tri-TSP | 20 | $(20, 20, 20)$ | $(0, 0)$ |
| | 50 | $(35, 35, 35)$ | $(0, 0)$ |
| | 100 | $(65, 65, 65)$ | $(0, 0)$ |

## G  Instance Augmentation

To further improve the performance of POCCO at the inference stage, we apply the instance augmentation proposed in [30], which is also used in PMOCO [38]. The rationale of instance augmentation is that an instance of Euclidean VRPs can be transformed into different ones that share the same optimal solution, e.g., by flipping coordinates for all nodes in an instance. Given a coordinate $(x, y)$ in a VRP, there are eight simple transformations, i.e., $(x', y') = (x, y); (y, x); (x, 1-y); (y, 1-x); (1-x, y); (1-y, x); (1-x, 1-y); (1-y, 1-x)$. In our paper, we adopt these transformations for each objective, respectively. Hence, we could have 8 transformations for Bi-CVRP (since there is only one coordinate for each node), $8^2 = 64$ transformations for Bi-TSP, and $8^3 = 512$ transformations for Tri-TSP, respectively.

## H  Impact of Neutralizing Non-Dominated Pairs

Our method adopts a decomposition-based framework, where the MOCOP is divided into a set of scalarized subproblems, each associated with a weight vector. During training, the goal is to find high-quality solutions with respect to each subproblem, rather than globally exploring the entire Pareto front at once. In this context, preference learning operates at the subproblem level. We acknowledge, however, that sampled solutions may occasionally be mutually non-dominated in the original objective space. To investigate the effect of this scenario, we conducted an additional experiment where preference signals for such non-dominated pairs were set to 0, effectively treating them as equivalent. The results, shown in Table 5, indicate that this modification (NDS) significantly degrades model performance across all Bi-TSP instances.

This observation suggests that introducing indifference signals for non-dominated pairs hinders the learning process. One reason is that it reduces the amount of effective preference supervision per subproblem, especially in early training stages when many solutions are of similar quality. In contrast, using scalarized values provides a consistent and differentiable training signal aligned with the decomposition objective. Therefore, although scalarized comparisons may not reflect global Pareto dominance in every case, they remain well-justified and necessary within the decomposition-based learning paradigm, and do not appear to bias the overall Pareto front approximation based on our empirical findings.

Table 5: Performance on Bi-TSP instances when neutralizing non-dominated pairs.

| Problem Size | POCCO-W HV | NDS HV |
|---|---|---|
| Bi-TSP20 | 0.6275 | 0.5516 |
| Bi-TSP50 | 0.6411 | 0.5160 |
| Bi-TSP100 | 0.7055 | 0.5601 |
| Bi-TSP150 | 0.7033 | 0.5494 |
| Bi-TSP200 | 0.7371 | 0.5809 |

## I  Detailed Results on Benchmark Instances

The detailed out-of-distribution generalization results are presented in Table 6, further confirming the exceptional generalization ability of our POCCO.

## J  Experimental Results on the Larger Problem Sizes

The results on Bi-TSP150/200, summarized in Table 7, show that POCCO-W consistently achieves the best generalization performance compared to all neural baselines and classical MOEAs.

Furthermore, to demonstrate scalability, we include experimental results on even larger problems, including Bi-TSP with 300/500/1000 nodes and MOKP with 300/500/1000 items. These results are summarized in Table 8 and Table 9. As shown, POCCO-W consistently outperforms the baseline WE-CA in most large-scale cases.

## K  Comparison with DPO

We have added a comparative experiment between our preference learning (PL) method and the recent DPO objective, as shown in Table 10. Since DPO requires two models, i.e., a policy model and a reference model, we use the same architecture and initialization for both, with the reference model updated from the previous epoch. This setup significantly increases training time for DPO (122h vs. 36h for POCCO-W).

As shown in Table 10, while DPO achieves slightly better performance on the smallest instance (MOTSP20), it is consistently outperformed by POCCO-W on larger instances (MOTSP50–200). These results suggest that our PL method offers a more favorable trade-off in terms of both training efficiency and solution quality, especially for large-scale problems.

Table 6: Performance on KroAB Instances

| | KroAB100 | | | KroAB150 | | | KroAB200 | | |
|---|---|---|---|---|---|---|---|---|---|
| Method | HV | Gap | Time | HV | Gap | Time | HV | Gap | Time |
| WS-LKH | **0.7022** | -0.23% | 2.3m | **0.7017** | -0.59% | 4.0m | **0.7430** | -0.83% | 5.6m |
| MOEA/D | 0.6836 | 2.43% | 5.8m | 0.6710 | 3.81% | 7.1m | 0.7106 | 3.57% | 7.3m |
| NSGA-II | 0.6676 | 4.71% | 7.0m | 0.6552 | 6.08% | 7.9m | 0.7011 | 4.86% | 8.4m |
| MOGLS | 0.6817 | 2.70% | 52m | 0.6671 | 4.37% | 1.3h | 0.7083 | 3.88% | 1.6h |
| PPLS/D-C | 0.6785 | 3.15% | 38m | 0.6659 | 4.54% | 1.4h | 0.7100 | 3.65% | 3.8h |
| DRL-MOA | 0.6903 | 1.47% | 10s | 0.6794 | 2.61% | 12s | 0.7185 | 2.50% | 18s |
| MDRL | 0.6881 | 1.78% | 9s | 0.6831 | 2.08% | 11s | 0.7209 | 2.17% | 16s |
| EMNH | 0.6900 | 1.51% | 9s | 0.6832 | 2.06% | 11s | 0.7217 | 2.06% | 16s |
| PMOCO | 0.6878 | 1.83% | 9s | 0.6819 | 2.25% | 12s | 0.7193 | 2.39% | 17s |
| CNH | 0.6947 | 0.84% | 16s | 0.6892 | 1.20% | 19s | 0.7250 | 1.61% | 22s |
| **POCCO-C** | 0.6965 | 0.59% | 30s | 0.6925 | 0.73% | 40s | 0.7302 | 0.91% | 50s |
| WE-CA | 0.6948 | 0.83% | 9s | 0.6924 | 0.75% | 12s | 0.7317 | 0.71% | 16s |
| **POCCO-W** | 0.6981 | 0.36% | 20s | 0.6946 | 0.43% | 31s | 0.7345 | 0.33% | 40s |
| MDRL-Aug | 0.6950 | 0.80% | 10s | 0.6890 | 1.23% | 16s | 0.7261 | 1.47% | 25s |
| EMNH-Aug | 0.6958 | 0.69% | 10s | 0.6892 | 1.20% | 16s | 0.7270 | 1.34% | 25s |
| PMOCO-Aug | 0.6937 | 0.98% | 11s | 0.6886 | 1.29% | 18s | 0.7251 | 1.60% | 30s |
| CNH-Aug | 0.6980 | 0.37% | 17s | 0.6938 | 0.54% | 26s | 0.7303 | 0.90% | 37s |
| **POCCO-C-Aug** | 0.6999 | 0.10% | 32s | 0.6959 | 0.24% | 48s | 0.7334 | 0.47% | 1.1m |
| WE-CA-Aug | 0.6990 | 0.23% | 10s | 0.6957 | 0.27% | 20s | 0.7349 | 0.27% | 31s |
| **POCCO-W-Aug** | 0.7006 | 0.00% | 22s | 0.6976 | 0.00% | 39s | 0.7369 | 0.00% | 59s |

Table 7: Performance on Bi-TSP150 and Bi-TSP200 Instances

| | Bi-TSP150 | | | Bi-TSP200 | | |
|---|---|---|---|---|---|---|
| Method | HV | Gap | Time | HV | Gap | Time |
| WS-LKH | **0.7149** | –1.23% | 13h | **0.7490** | –1.23% | 22h |
| MOEA/D | 0.6809 | 3.58% | 2.4h | 0.7139 | 3.51% | 2.7h |
| NSGA-II | 0.6659 | 5.71% | 6.8h | 0.7045 | 4.78% | 6.9h |
| MOGLS | 0.6768 | 4.16% | 22h | 0.7114 | 3.85% | 38h |
| PPLS/D-C | 0.6784 | 3.94% | 21h | 0.7106 | 3.96% | 32h |
| DRL-MOA | 0.6901 | 2.28% | 36s | 0.7219 | 2.43% | 1.2m |
| MDRL | 0.6922 | 1.98% | 36s | 0.7251 | 2.00% | 1.1m |
| EMNH | 0.6930 | 1.87% | 37s | 0.7260 | 1.88% | 1.1m |
| PMOCO | 0.6910 | 2.15% | 42s | 0.7231 | 2.27% | 1.3m |
| CNH | 0.6985 | 1.09% | 50s | 0.7292 | 1.45% | 1.4m |
| **POCCO-C** | 0.7011 | 0.72% | 1.5m | 0.7333 | 0.89% | 2.5m |
| WE-CA | 0.7008 | 0.76% | 45s | 0.7346 | 0.72% | 1.3m |
| **POCCO-W** | 0.7033 | 0.41% | 1.4m | 0.7371 | 0.38% | 2.4m |
| MDRL-Aug | 0.6976 | 1.22% | 37m | 0.7299 | 1.35% | 1.1h |
| EMNH-Aug | 0.6983 | 1.12% | 39m | 0.7307 | 1.24% | 1.1h |
| PMOCO-Aug | 0.6967 | 1.35% | 40m | 0.7283 | 1.57% | 1.2h |
| CNH-Aug | 0.7025 | 0.52% | 41m | 0.7343 | 0.76% | 1.2h |
| **POCCO-C-Aug** | 0.7043 | 0.27% | 55m | 0.7366 | 0.45% | 1.5h |
| WE-CA-Aug | 0.7044 | 0.25% | 42m | 0.7381 | 0.24% | 1.2h |
| **POCCO-W-Aug** | 0.7062 | 0.00% | 45m | 0.7399 | 0.00% | 1.4h |

# L   Hyperparameter Study

**Effects of the $\beta$.** Fig.5 shows how the temperature parameter $\beta$ in the preference learning loss affects performance (HV) across four benchmark tasks. A moderate value of $\beta$ consistently yields the best results. For the three bi-objective problems (Bi-TSP100, MOCVRP100, Bi-KP100), performance peaks around $\beta = 3.5$; larger or smaller values provide no additional benefit, and in Bi-KP100, an overly large $\beta = 5$ sharply degrades HV. For the tri-objective problem (Tri-TSP100), performance improves up to $\beta = 4.5$ and then declines slightly. Overall, these trends justify the default settings adopted in our experiments: $\beta = 3.5$ for bi-objective tasks and $\beta = 4.5$ for tri-objective tasks.

Table 8: HV on large Bi-TSP instances.

| Problem Size | WE-CA | POCCO-W |
|---|---|---|
| Bi-TSP300 | 0.7441 | **0.7458** |
| Bi-TSP500 | 0.7476 | **0.7592** |
| Bi-TSP1000 | 0.7186 | **0.7400** |

Table 9: HV on large MOKP instances.

| Problem Size | WE-CA | POCCO-W |
|---|---|---|
| Bi-KP300 | **0.6000** | 0.5947 |
| Bi-KP500 | 0.4244 | **0.5658** |
| Bi-KP1000 | 0.2439 | **0.8408** |

Table 10: HV of different PL methods on Bi-TSP instances.

| Problem Size | POCCO-W | DPO |
|---|---|---|
| Bi-TSP20 | 0.6275 | **0.6276** |
| Bi-TSP50 | **0.6411** | 0.6400 |
| Bi-TSP100 | **0.7055** | 0.7030 |
| Bi-TSP150 | **0.7033** | 0.6998 |
| Bi-TSP200 | **0.7371** | 0.7331 |

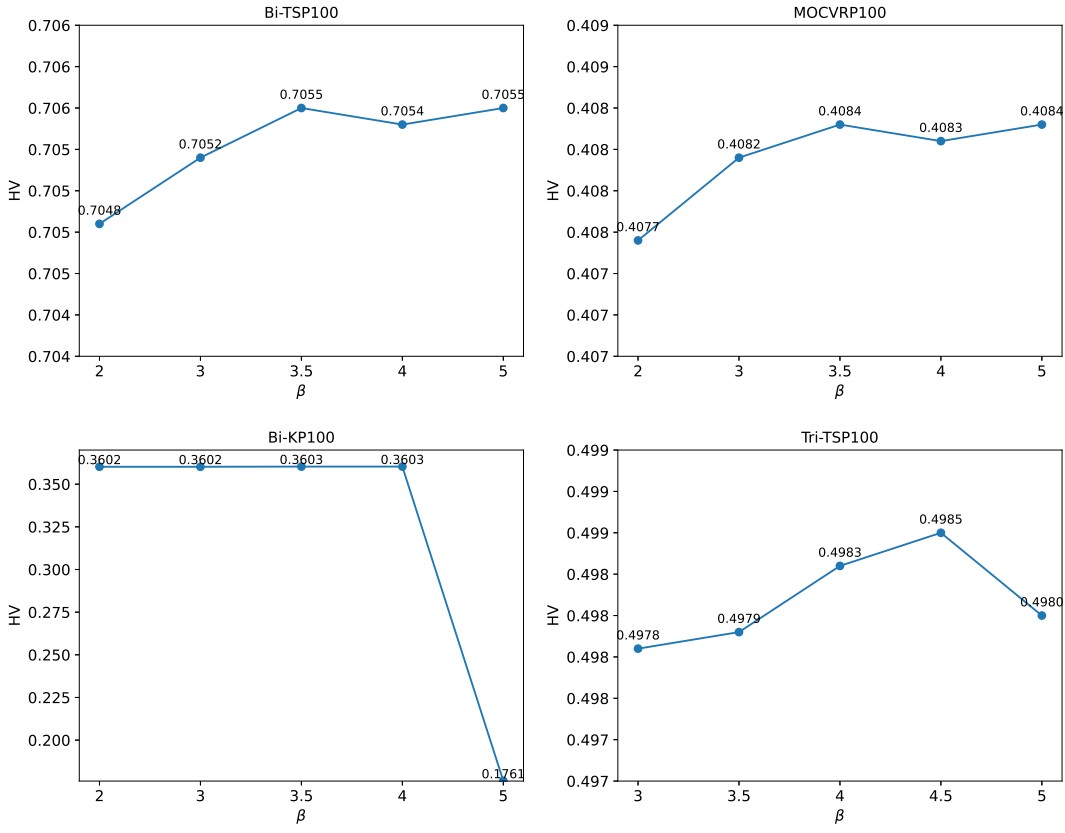

Figure 5: Effectis of the $\beta$.

**Effects of the number of CCO layers.** The CCO block sits in the decoder, whose sequence length T grows with problem size. Each additional CCO layer, therefore, adds a full set of gating and expert computations at every decoding step. Hence, stacking too many CCO layers can inflate the runtime disproportionately. Table 11 reports the trade-off. Using two CCO layers yields the highest HV on both Bi-TSP100 and Bi-TSP200, but increases inference time by roughly 60% and 130 %, respectively. A third layer further slows inference while slightly reducing HV. We therefore adopt a single-layer CCO as the default, which preserves most of the performance gain while keeping computation modest. For larger instances or latency-sensitive applications, the one-layer setting offers a favorable balance between solution quality and speed.

Table 11: Effects of the number of CCO block layers.

| Method | Bi-TSP100 | | Bi-TSP200 | |
|---|---|---|---|---|
| | HV | Time | HV | Time |
| POCCO-W | 0.7055 | 36 s | 0.7371 | 2.4 m |
| 2 CCO layers | 0.7058 | 59 s | 0.7379 | 5.6 m |
| 3 CCO layers | 0.7049 | 85 s | 0.7352 | 7.9 m |

**Effects of the number of experts.** We have added an ablation study to analyze the impact of the number of experts in Table 12. Specifically, POCCO-W (5E) refers to the original model presented in the paper, which includes 4 feedforward (FF) experts and 1 parameter-free identity (ID) expert. We additionally evaluate four variants:

1. POCCO-W (3E): 2 FF + 1 ID;

2. POCCO-W (9E): 8 FF + 1 ID;

3. POCCO-W (9E_2D): 8 FF + 1 ID, trained on twice the amount of data;

4. POCCO-W (17E): 16 FF + 1 ID.

All variants are trained under the same settings as POCCO-W (5E) for a fair comparison, except for POCCO-W (9E_2D), which is trained on more data.

As shown in Table 12, all POCCO-W variants outperform the backbone model WE-CA, confirming the effectiveness of the expert-based architecture. Moreover, POCCO-W (3E) and POCCO-W (5E) achieve better overall performance than POCCO-W (9E) and POCCO-W (17E), the latter of which requires additional data scaling to realize performance gains. To strike a better balance between computational cost and solution quality, we select POCCO-W (5E) as our default model.

Table 12: Effects of the number of experts.

| Problem Size | WE-CA | POCCO-W (3E) | POCCO-W (5E) | Exp_num (9E) | Exp_num (9E_2D) | Exp_num (17E) |
|---|---|---|---|---|---|---|
| Bi-TSP20 | 0.6270 | **0.6275** | **0.6275** | **0.6275** | **0.6275** | **0.6275** |
| Bi-TSP50 | 0.6392 | 0.6410 | **0.6411** | 0.6408 | **0.6411** | 0.6408 |
| Bi-TSP100 | 0.7034 | **0.7057** | 0.7055 | 0.7050 | 0.7053 | 0.7050 |
| Bi-TSP150 | 0.7008 | **0.7035** | 0.7033 | 0.7026 | 0.7031 | 0.7027 |
| Bo-TSP200 | 0.7346 | 0.7363 | **0.7371** | 0.7364 | 0.7370 | 0.7361 |

**Effects of** Top $k$**.** Fig.6 plots validation HV on Bi-TSP100 for $k \in \{1, 2, 3\}$. Selecting two experts per token ($k$=2) converges fastest and attains the highest final HV. A single expert ($k$=1) limits model capacity, leading to slower early progress and a slightly lower plateau. Choosing three experts ($k$=3) increases computation relative to $k$=2 yet offers no benefit and even marginally reduces HV, likely because the additional expert dilutes specialization and weakens sparsity. We therefore adopt $k$=2 as the default, balancing performance and efficiency.

# M    Experimental Results of Scalarized Subproblems

To assess subproblem optimality, we report scalarized objective values for three representative weight vectors $\lambda = (1, 0), (0.5, 0.5), (0, 1)$ on Bi-TSP100 (Table 13). We also compare against single-objective solvers LKH and POMO, each tuned to the corresponding subproblem. Among

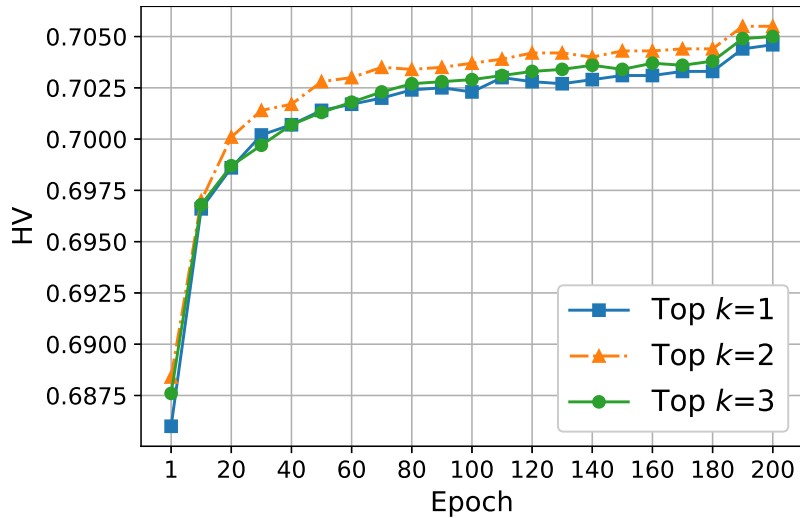

Figure 6: Effects of Top $k$.

neural MOCOP methods, POCCO-W attains the smallest optimality gaps across all weight settings. Ablating CCO (POCCO-Aug w/o CCO) increases optimality gaps across all settings, and eliminating preference learning as well (WE-CA-Aug) further degrades performance. Notably, POCCO-W-Aug even outperforms POMO-Aug on the subproblem with $\lambda = (0, 1)$.

Table 13: Performance comparison under different weight settings $(\omega_1, \omega_2)$.

| Method | $\lambda = (1, 0)$ | | $\lambda = (0.5, 0.5)$ | | $\lambda = (0, 1)$ | |
| --- | --- | --- | --- | --- | --- | --- |
| | Obj | Gap | Obj | Gap | Obj | Gap |
| WS-LKH | 7.7632 | 0.00% | 17.3094 | 0.00% | 7.7413 | 0.00% |
| POMO-Aug | 7.7659 | 0.03% | 17.4421 | 0.77% | 7.7716 | 0.39% |
| WE-CA-Aug | 7.8132 | 0.64% | 17.4994 | 1.10% | 7.7888 | 0.61% |
| POCCO-Aug w/o CCO | 7.7970 | 0.44% | 17.4629 | 0.89% | 7.7772 | 0.46% |
| POCCO-W-Aug | 7.7827 | 0.25% | 17.4460 | 0.79% | 7.7602 | 0.24% |

## N Comparison with Tchebycheff Scalarization

Intuitively, our POCCO framework is applicable to any MOCOP where the objectives can be scalarized using decomposition-based techniques, including weighted-sum (WS), Tchebycheff (TCH), and penalty-based boundary intersection (PBI). This is because the preference signal between two solutions is determined solely by the ordering of their scalarized objective values, regardless of the specific scalarization method used.

To empirically demonstrate this generality, we added an experiment comparing POCCO-W trained with WS and TCH scalarization methods. As shown in Table 14, WS consistently outperforms TCH on Bi-TSP50 to Bi-TSP200, indicating that WS offers more robust performance across different problem sizes. This observation is also consistent with findings in [6], which report that WS—despite its simplicity—often outperforms TCH in the studied MOCOP setting. In general, however, WE-CA (TCH) performs worse than WE-CA (WS), which achieves a score of 0.6392, and is further outperformed by POCCO-W (TCH), which achieves 0.6395 on Bi-TSP50.

## O GPU Memory Usage

We present a comparison of GPU memory usage between POCCO-W (ours), the backbone model (WE-CA), and CNH in Table 15. All models are trained across problem instances with size $n = 50$. As shown in Table 15, POCCO-W incurs higher GPU memory usage, approximately doubling that of

Table 14: HV of Different Decomposition Approaches on Bi-TSP Instances.

| Problem Size | WE-CA (WS) | POCCO-W (WS) | POCCO-W (TCH) |
|---|---|---|---|
| Bi-TSP50 | 0.6392 | 0.6411 | 0.6395 |
| Bi-TSP100 | 0.7034 | 0.7055 | 0.7031 |
| Bi-TSP150 | 0.7008 | 0.7033 | 0.6998 |
| Bi-TSP200 | 0.7346 | 0.7371 | 0.7329 |

Table 15: Comparison of GPU Memory Usage.

| Problem Size | WE-CA | CNH | POCCO-W |
|---|---|---|---|
| Bi-TSP50 | 1051 MB | 1077 MB | 1849 MB |
| MOCVRP50 | 1790 MB | 1501 MB | 4522 MB |
| Bi-KP50 | 916 MB | 916 MB | 2417 MB |
| Tri-TSP50 | 1052 MB | 1077 MB | 2383 MB |

the baselines WE-CA and CNH. However, this increased resource cost is accompanied by significant improvements in solution quality, as demonstrated in our experimental results.

## P   Summary of Decomposition-based Neural MOCOP Solvers

Table 16 compares key features of decomposition-based neural MOCOP solvers. POCCO, which establishes new SOTA performance, is a plug-and-play framework that augments any existing solver. It inserts a CCO block that learns a diverse ensemble of policies, adding parameters yet surpassing prior methods. Crucially, each subproblem activates only two experts (one of which may be a parameter-free identity expert), so the extra computational load remains minimal. POCCO also employs a pairwise preference learning approach that further boosts performance without introducing additional parameters.

Table 16: Summary of the decomposition-based neural MOCOP solvers.

| Method | Learning method | Paradigm | #Parameters |
|---|---|---|---|
| DRL-MOA | Transfer learning+RL | one-to-one | 133.37M |
| MDRL | Meta learning+RL | one-to-one | 133.37M |
| EMNH | Meta learning+RL | one-to-one | 133.37M |
| PMOCO | RL | one-to-many | 1.50M |
| CNH | RL | one-to-many | 1.63M |
| **POCCO-C** | Preference learning | one-to-many | 2.16M |
| WE-CA | RL | one-to-many | 1.47M |
| **POCCO-W** | Preference learning | one-to-many | 2.00M |

## Q   Broader Impacts

POCCO offers several positive societal impacts. By dynamically routing computation and learning from preference signals, it accelerates multi-objective decision-making in logistics, manufacturing, and energy planning, reducing costly trial-and-error loops and boosting operational efficiency. Its conditional-computation design activates only the required network capacity for each subproblem, cutting FLOP counts and energy consumption relative to dense models of comparable accuracy. Finally, by encouraging exploration and adaptive capacity allocation, POCCO broadens the applicability of neural solvers in complex optimization tasks, advancing both AI and operations research and enabling practitioners to tackle larger, real-world problems with fewer computational resources.

# R  Licenses for Existing Assets

The used assets in this work are listed in Table 17. Where applicable, we reference publicly available implementations for evaluation or reproduction purposes. Our source code is released under the MIT License.

Table 17: Used assets, licenses, and their usage.

| Type | Asset | License | Usage |
|---|---|---|---|
| Code | LKH [20] | Available for academic use | Evaluation |
| | DRL-MOA [34] | No license (assumed all rights reserved) | Evaluation |
| | MDRL [67] | No license (assumed all rights reserved) | Evaluation |
| | EMNH [7] | No license (assumed all rights reserved) | Evaluation |
| | PMOCO [38] | MIT License | Evaluation |
| | CNH [14] | MIT License | Revision |
| | WE-CA [6] | MIT License | Revision |
| Datasets | Chen et al.[6] | MIT License | Evaluation |

