# OpenReview forum: "Preference-Driven Multi-Objective Combinatorial Optimization with Conditional Computation"
_NeurIPS.cc/2025/Conference — NeurIPS 2025 poster_

### Official Review · Reviewer_FKaN · 2025-06-23

**Clarity:** 3
**Significance:** 3
**Originality:** 3
**Rating:** 4
**Confidence:** 3

**Summary:**

This paper proposes a plug-and-play multi-objective reinforcement learning (RL) framework, POCCO. It consists of two major components.
(1) Conditional Computation Block (CCO): It is a sparse router that employs a gated network, combined with a Top-k operator, to enable the model to decide which experts to activate. This can ensure the width and depth while remaining efficient. This aims to improve the exploration over pure REINFORCE.
(2) Preference-driven Learning: This method is inspired by the Bradley-Terry likelihood, where the weighted sum of rewards is replaced with the average log-likelihood gap between two randomly selected complete solutions. This aims to lower the gradient variance.

The experiments are done on bi/tri-objective TSP, CVRP, and Knapsack problems. The results look promising, and the backbone is improved when enhanced by their method.

**Questions:**

1. It would be great if you had more experimental results on some more complex problems. The backbones are already performing quite well for now, so the improvements seem minor.

2. For comparison, you have shown that with PL, your convergence is significantly faster than with pure REINFORCE. Then why not set the solving time the same for both the backbone and backbone + your method? This seems to be a better comparison.

**Ethical Concerns:**

["NO or VERY MINOR ethics concerns only"]

**Final Justification:**

All my questions are resolved and I will keep my score unchanged.

**Limitations:**

Yes.

**Quality:**

3

**Strengths And Weaknesses:**

Strengths:
(1) The method is a plug-and-play framework, and the two mechanisms are simple and elegant. So it would be easy to adapt to most multi-objective RL frameworks.

Weaknesses:
(1) The test problems are a bit "easy" since the backbone alone has already been very close to the optimal results. So the improvement seems minor. See more in the questions.
(2) Some experimental settings are questionable. See more in the questions.

---

> ### Author Rebuttal · Authors · 2025-07-31
>
> We sincerely appreciate the reviewer for providing positive and constructive comments. Here are our detailed responses to your comments, where W denotes Weakness (W1-W2) and Q denotes Question (Q1-Q2).
>
> **W1 & Q1: The test problems are a bit "easy" since the backbone alone has already been very close to the optimal results. So the improvement seems minor. It would be great if you had more experimental results on some more complex problems.**
>
> We acknowledge that the benchmark problems used in our main experiments—Bi-TSP, Tri-TSP, Bi-CVRP, and Bi-KP—are widely adopted in prior neural MOCOP work [1–5], which facilitates fair comparison with existing baselines. However, we agree that these instances may not fully reflect the challenges posed by real-world constraints.
>
> Following the reviewer’s suggestion to address the applicability of our method under more complex constraints, we conducted an additional experiment on the multi-objective capacitated vehicle routing problem with time windows (MOCVRPTW), as summarized in Table G. In the MOCVRPTW, each customer node $v_i$ is associated with a time window $[e_i, l_i]$ and a service time $s_i$[6]. A vehicle must start serving customer $v_i$ in the time slot from $e_i$ to $l_i$.  If the vehicle arrives earlier than $e_i$, it has to wait until $e_i$. All vehicles must return to the depot $v_0$ no later than $l_0$. This study considers three objectives: minimizing the total travel distance, minimizing the number of routes, and minimizing the average route length per customer served. As shown in Table G, POCCO-W consistently outperforms WE-CA on MOCVRPTW tasks as well.
>
> **Table G: HV on MOCVRPTW Instances.**
> | Problem Size | WE-CA HV | POCCO-W HV |
> |--------------|:--------:|:----------:|
> | MOCVRPTW20    | 0.5465   | **0.5496** |
> | MOCVRPTW50    | 0.5499   | **0.5576** |
> | MOCVRPTW100   | 0.5331   | **0.5375** |
>
> **W2 & Q2: Why not set the solving time the same for both the backbone and backbone + your method? This seems to be a better comparison.**
>
> Thanks for your suggestion. We have added a new experiment to compare POCCO-W with the backbone method (WE-CA) under the same solving time constraint. Specifically, we limit each instance to a 10-second solution sampling budget for both methods and evaluate on the same test datasets used in the main paper (200 instances per problem setting). As shown in Table M, POCCO-W consistently outperforms WE-CA across all benchmark problems, even when both are given exactly the same solving time. We will include a comprehensive comparison under the same solving time in the final version of the paper.
>
> **Table M: HV of POCCO-W and WE-CA Under Equal Solving Time (10s per instance).**
> | Problem     | WE-CA  | POCCO-W  |
> |-------------|:--------------------:|:----------------------:|
> | Bi-TSP100   | 0.7049               | **0.7071**                 |
> | MOCVRP100   | 0.4076               |  **0.4090**               |
> | Bi-KP100    | 0.4531               | **0.4532**                |
> | Tri-TSP100  | 0.4999               |  **0.5022**                |
>
> [1] Pareto set learning for neural multi-objective combinatorial optimization, in ICLR 2022.
>
> [2] Neural multi-objective combinatorial optimization with diversity enhancement, in NeurIPS 2023.
>
> [3] Conditional neural heuristic for multiobjective vehicle routing problems, TNNLS, 2024.
>
> [4] Collaborative deep reinforcement learning for solving multi-objective vehicle routing problems, in AAMAS 2024.
>
> [5] Rethinking neural multi-objective combinatorial optimization via neat weight embedding, in ICLR 2025.
>
> [6] MVMoE: Multi‑Task Vehicle Routing Solver with Mixture‑of‑Experts, in ICML 2024.

---

> ### Author Response · Authors · 2025-08-07
>
> Dear Reviewer `FKaN`,
>
> Thank you for the time and effort you have devoted to reviewing our paper!
>
> With the discussion period now extended, we would welcome the opportunity to clarify any remaining questions or concerns you might have. Please let us know if there is any aspect of the work you would like us to elaborate on.
>
> We look forward to your reply and greatly appreciate your feedback.
>
> Best regards,
> Authors

---

### Official Review · Reviewer_kHSo · 2025-07-01

**Clarity:** 2
**Significance:** 3
**Originality:** 3
**Rating:** 5
**Confidence:** 3

**Summary:**

This paper proposes POCCO, a preference-driven multi-objective combinatorial optimization framework with conditional computation, which achieves performance improvements on MOCOPs. While the approach is innovative and empirically validated, several issues remain clarification.

**Questions:**

- Does the preference learning approach offer theoretical guarantees on better convergence or exploration efficiency compared to traditional REINFORCE?
- Are the experimental results still valid for larger-scale problem instances? What are the primary factors contributing to this limitation
- Does the introduction of multiple FF experts and dynamic routing significantly increase computational cost? Considering the potential need to handle more objectives, how do the authors envision scaling POCCO efficiently to accommodate high-dimensional objective spaces?
- Can the process of generating preference pairs introduce bias?
The paper reports strong out-of-distribution generalization performance. Is the claimed generalization ability dependent on the training distribution?

**Ethical Concerns:**

["NO or VERY MINOR ethics concerns only"]

**Final Justification:**

The authors have addressed my concerns.

**Limitations:**

yes

**Quality:**

3

**Strengths And Weaknesses:**

Strengths:
- The proposed approach is technically sound.
- The paper introduces a CCO in the decoder that dynamically routes subproblems through specialized feed-forward or identity experts based on context.
- The paper proposes a preference-driven optimization algorithm inspired by the Bradley–Terry model, which learns from pairwise solution comparisons rather than absolute rewards.
- The proposed POCCO is evaluated on several multi-objective combinatorial optimization benchmarks.


Weaknesses:
- While related work is mentioned in the introduction and appendix, the paper lacks a dedicated literature review section. This makes it difficult to understand the development trajectory of existing MOCOP methods and POCCO’s positioning within the field.
- The model complexity and computational cost are not clearly addressed, especially with the inclusion of multiple feed-forward experts and dynamic routing. Further clarification on training and inference efficiency—particularly in terms of parameter count, runtime, and scalability—is needed.
- The concept of “Preference Learning” is vaguely defined.

---

> ### Author Rebuttal · Authors · 2025-07-31
>
> We greatly appreciate the reviewer for providing constructive comments. Here are our detailed responses to your comments, where W denotes Weakness (W1-W3) and Q denotes Question (Q1-Q4).
>
> **W1: Lacks a dedicated related work section.**
>
> We acknowledge the missing related work section and will move relevant content from the appendix into the main text. A comprehensive literature review will be added in the final version to clarify the development of MOCOP methods and POCCO’s position.
>
> **W2 & Q3: Clarify training/inference efficiency and scalability. Does CCO add significant cost, and how does POCCO scale to many-objective problems?**
>
> We conducted an additional set of experiments comparing several neural MOCOP solvers in terms of model size, training time, GPU memory usage (on Bi-TSP50), and solution quality (HV). As shown in Tables 1–2 and Table H, POCCO-W incurs moderately higher training and inference overhead than CNH and WE-CA, but consistently achieves better performance. We believe this represents a reasonable trade-off between model complexity and performance.
>
> Specifically, the additional parameters in POCCO-W primarily come from four FF experts (totaling 526,848 parameters) and a lightweight router module (1,280 parameters). It is worth noting that during inference, each subproblem activates only two experts, one of which may be the parameter-free ID expert. This design helps keep the computational overhead manageable.
>
>
> **Table H: Overhead of Different Neural MOCOP solvers.**
> | Method    | Parameters | Training Time| GPU Usage (Bi-TSP50)|HV (Bi-TSP50)|
> |-----------|:-----------:|:-------------:|:-----------:|:-----------:|
> | CNH       | 1.63M       | 17h |1077 MB | 0.6387 |
>  WE-CA     | 1.47M       | 11h |1051 MB | 0.6392 |
> | POCCO-W   | 2.00M       | 36h |1849 MB  | 0.6411 |
>
> As for scalability to high-dimensional objective spaces, POCCO is designed within a decomposition-based framework where each training subproblem is defined by a scalarization (e.g., weighted-sum or Tchebycheff). This means that the model does not directly operate in the full objective space, but instead solves scalarized subproblems, allowing it to scale independently of the number of objectives.
>
> Nevertheless, we recognize that higher-dimensional Pareto fronts may require denser sampling of preference vectors, which could increase training cost. To address this, POCCO can be efficiently extended using the following strategies:
> * **Router conditioning:** The routing network can be conditioned on compressed representations of the weight vector (e.g., via projection or learned embeddings), avoiding a linear blowup in routing complexity.
> * **Expert reuse:** Experts can be shared across subsets of objectives, enabling parameter reuse across related regions of the Pareto front.
> * **Hierarchical routing:** For very high-dimensional cases, hierarchical or sparse gating strategies can further reduce expert activation and computational load.
>
> **W3: “Preference Learning” is vaguely defined.**
>
> Thank you for the comment. We acknowledge that the definition of “Preference Learning” (PL) in the current version may not be sufficiently clear. In our context, PL refers to a learning paradigm where the model is trained to prefer one solution over another based on their scalarized objective values under a given weight vector. Specifically, instead of learning from scalar rewards (as in RL), the model receives pairwise supervision derived from the relative quality of two sampled solutions. This preference signal is then used to guide policy updates through a ranking-based loss.
>
> We will revise the manuscript to include a more precise and formal definition of PL at its first mention and clarify its distinction from reinforcement learning approaches in both methodology and learning dynamics.
>
>
> **Q1: Does preference learning offer theoretical guarantees over REINFORCE?**
>
> The majority of the work [1–2] within the neural MOCOP literature focused on the empirical side, making the provision of a solid theoretical analysis within the tight rebuttal period a non-trivial challenge. Here, we try to give an in-depth empirical analysis concerning the effects of preference learning (PL) method, with the aim of offering valuable insights for future research.
>
> Compared to standard REINFORCE-based reinforcement learning, our PL approach offers significant practical advantages. Instead of relying on high-variance reward signals (as in REINFORCE), PL compares solutions under the same weight vector, providing a low-variance signal. We provide concrete evidence of this in Table D, which reports the average gradient variance in the first five batches of training POCCO-W under both RL and PL frameworks. Across three MOTSP settings, PL consistently exhibits two to four orders of magnitude lower variance, leading to faster convergence and more stable optimization dynamics.
>
> **Table D: Gradient variance in REINFORCE vs. preference learning.**
> | Problem Size | Batch | RL Variance | PL Variance |
> |--------------|:-----:|:----------------:|:----------------:|
> | Bi-TSP20      |   1   |     0.054648     |     0.000314     |
> |              |   2   |     0.039951     |     0.000252     |
> |              |   3   |     0.019038     |     0.000191     |
> |              |   4   |     0.009093     |     0.000115     |
> |              |   5   |     0.010742     |     0.000104     |
> | Bi-TSP50      |   1   |     0.474784     |     0.000142     |
> |              |   2   |     0.220530     |     0.000077     |
> |              |   3   |     0.140275     |     0.000048     |
> |              |   4   |     0.092114     |     0.000040     |
> |              |   5   |     0.065286     |     0.000031     |
> | Bi-TSP100     |   1   |    10.124832     |     0.000059     |
> |              |   2   |     6.355499     |     0.000039     |
> |              |   3   |     3.461700     |     0.000032     |
> |              |   4   |     1.755804     |     0.000021     |
> |              |   5   |     1.111246     |     0.000014     |
>
>
> **Q2: Do results generalize to larger-scale instances? What limits scalability?**
>
> To evaluate the out-of-distribution generalization ability of our method, we report additional results on larger Bi-TSP instances (150 and 200 nodes) as well as benchmark instances (KroAB150 and KroAB200) in Appendix H and I, respectively. Furthermore, to demonstrate scalability, we include new experimental results on even larger problems including Bi-TSP with 300/500/1000 nodes and MOKP with 300/500/1000 items. These results are summarized in Table E and Table F. Due to computational constraints, we generate 5 random instances for each setting. As shown, POCCO-W consistently outperforms the baseline WE-CA in most large-scale cases.
>
> **Table E: HV on large Bi-TSP instances.**
> | Problem Size | WE-CA HV | POCCO-W HV |
> |--------------|:--------:|:----------:|
> | Bi-TSP300     | 0.7441   | **0.7458**     |
> | Bi-TSP500     | 0.7476   | **0.7592**     |
> | Bi-TSP1000    | 0.7186   | **0.7400**     |
>
> **Table F: HV on large MOKP instances.**
> | Problem Size | WE-CA HV | POCCO-W HV |
> |--------------|:--------:|:----------:|
> | Bi-KP300      | **0.6000**   | 0.5947     |
> | Bi-KP500      | 0.4244   | **0.5658**     |
> | Bi-KP1000     | 0.2439   | **0.8408**     |
>
> We acknowledge, however, that the performance of our method may degrade when applied to extremely large-scale problems (e.g., 10,000-node MOTSP). This limitation is primarily due to several factors:
> * **Decoder length and exposure bias:** In very long decoding sequences, small prediction errors accumulate, reducing solution quality.
> * **Router generalization gap:** The context distribution in extremely large problems deviates from training data, making expert routing less effective.
> * **Expert capacity saturation:** A fixed number of experts may be insufficient to model the growing diversity of subproblem structures.
>
> We consider scaling to ultra-large MOCOPs a valuable direction for future work and plan to explore hierarchical decoding, adaptive expert allocation, and curriculum-based training to further extend POCCO’s applicability.
>
> **Q4: Does preference pair generation introduce bias? Is generalization tied to training distribution?**
>
> Our method generates preference pairs by sampling two candidate solutions under the same weight vector and comparing their scalarized objective values. This process is inherently tied to the decomposition-based formulation of multi-objective optimization, where each weight vector defines a specific subproblem. Thus, the preference signal is locally defined within each subproblem and does not rely on global dominance relationships.
>
> Regarding potential bias: while it is true that the quality of the preference supervision depends on the candidate solution pool (e.g., from early-stage policies), we mitigate this by continuously sampling pairs from the evolving policy during training. This dynamic sampling ensures that preference pairs reflect the model’s current solution space and are not biased toward a fixed subset of the Pareto front.
>
> As for generalization, Figs. 3-4, Appendices H–I, and the large-scale experiments in Q2 demonstrate strong out-of-distribution generalization across problem sizes. Our model is trained on a range of problem sizes $n \in \[20, 21, \dots, 100\]$ (and $\[50, \dots, 200\]$ for Bi-KP), following [1-2], which showed that training across sizes leads to better generalization than training on a fixed size. We believe this range-based training enables the model to learn distributional patterns of subproblem structures, rather than overfitting to specific instance sizes. We will include an analysis of the effect of training size distribution in the final version to clarify this further.
>
> [1] Conditional neural heuristic for multiobjective vehicle routing problems, TNNLS, 2024.
>
> [2] Rethinking neural multi-objective combinatorial optimization via neat weight embedding, in ICLR 2025.

---

> > ### Comment · Reviewer_kHSo · 2025-08-05
> >
> > I thank the authors for the detailed answer. I raise my rating to 5.

---

> > > ### Author Response · Authors · 2025-08-05
> > >
> > > Thank you for your kind feedback and for updating the score. We truly appreciate your thoughtful comments, which helped us refine and strengthen our work.

---

### Official Review · Reviewer_CeFA · 2025-07-02

**Clarity:** 3
**Significance:** 2
**Originality:** 3
**Rating:** 4
**Confidence:** 3

**Summary:**

This paper proposes POCCO, a plug-and-play framework for Multi-Objective Combinatorial Optimization Problems (MOCOPs) using deep reinforcement learning. Unlike existing methods that rely on scalarized rewards and shared models across subproblems, POCCO introduces two key components: a conditional computation block that routes subproblems to specialized neural architectures, and a preference-driven optimization algorithm that leverages pairwise preferences.

**Questions:**

1. How is the routing mechanism in the CCO block learned? Specifically, how does the sparse gating network select between FF and ID experts, and what role does the Top-k operation play?

2. When sampled solutions are non-dominated, how does the scalarized value determine the winner in preference learning? Could this bias the Pareto front exploration?

3. The paper claims efficiency gains—could you break down the contributions of the CCO block and preference-driven training to the overall speedup?

**Ethical Concerns:**

["NO or VERY MINOR ethics concerns only"]

**Final Justification:**

The authors have addressed all my concerns.

**Limitations:**

yes

**Quality:**

3

**Strengths And Weaknesses:**

Strengths:
1. POCCO introduces a novel plug-and-play framework that effectively tackles key limitations of existing MOCOP methods, including insufficient exploration and limited model capacity.
2. The paper provides substantial experimental verification across multiple classic MOCOP benchmarks, exhibiting strong generalization capabilities and excellent performance even when faced with unseen problem scales and out-of-distribution instances.

Weaknesses:
1. The paper lacks theoretical support for its design choices. For instance, it’s unclear how the CCO block enhances exploration or representation learning, or how preference learning mitigates gradient variance in REINFORCE.
2. When generating preference pairs, the paper samples two candidate solutions from the policy and determines superiority based on the scalarized objective values. However, in complicated multi-objective settings, solutions can be mutually non-dominated.

---

> ### Author Rebuttal · Authors · 2025-07-31
>
> We appreciate your valuable comments.
>
> **W1: Lacks theoretical justification how CCO aids representation and how PL reduces gradient variance.**
>
> The majority of the work [1–5] within the neural multi-objective combinatorial optimization (MOCO) literature focused on the empirical side, making the provision of a solid theoretical analysis within the tight rebuttal period a non-trivial challenge. Here, we try to give an in-depth empirical analysis concerning the effects of the CCO module and preference learning method, with the aim of offering valuable insights for future research.
>
> The CCO block, positioned between the decoder’s attention and compatibility layers, enhances the model's representation capacity by dynamically selecting among multiple experts for each subproblem. The model can conditionally adapt its decoding behavior based on both context and objective preference. This modularity encourages expert specialization, enabling the decoder to capture a richer and more diverse set of mappings from inputs to actions. Since different weight vectors (in decomposition-based learning) induce different subproblem characteristics, the CCO router can learn to route similar subproblems to similar experts, effectively partitioning the representation space [6-7]. This facilitates both exploration across diverse sub-tasks and robustness to task shift, which is particularly beneficial in multi-objective optimization.
>
> Compared to standard REINFORCE-based reinforcement learning, our preference learning (PL) approach offers significant practical advantages. Instead of relying on high-variance reward signals (as in REINFORCE), PL compares solutions under the same weight vector, providing a low-variance signal. We provide concrete evidence of this in Table D, which reports the average gradient variance in the first five batches of training POCCO-W under both RL and PL frameworks. Across three MOTSP settings, PL consistently exhibits two to four orders of magnitude lower variance, leading to faster convergence and more stable optimization dynamics.
>
> **Table D: Gradient variance in REINFORCE vs. preference learning.**
> | Problem Size | Batch | RL Variance | PL Variance |
> |--------------|:-----:|:----------------:|:----------------:|
> | MOTSP20      |   1   |     0.054648     |     0.000314     |
> |              |   2   |     0.039951     |     0.000252     |
> |              |   3   |     0.019038     |     0.000191     |
> |              |   4   |     0.009093     |     0.000115     |
> |              |   5   |     0.010742     |     0.000104     |
> | MOTSP50      |   1   |     0.474784     |     0.000142     |
> |              |   2   |     0.220530     |     0.000077     |
> |              |   3   |     0.140275     |     0.000048     |
> |              |   4   |     0.092114     |     0.000040     |
> |              |   5   |     0.065286     |     0.000031     |
> | MOTSP100     |   1   |    10.124832     |     0.000059     |
> |              |   2   |     6.355499     |     0.000039     |
> |              |   3   |     3.461700     |     0.000032     |
> |              |   4   |     1.755804     |     0.000021     |
> |              |   5   |     1.111246     |     0.000014     |
>
>
>
> **W2 & Q2: How does PL handle non-dominated pairs, and could bias Pareto exploration?**
>
> Our method adopts a decomposition-based framework, where the MOCOP is divided into a set of scalarized subproblems, each associated with a weight vector. During training, the goal is to find high-quality solutions with respect to each subproblem, rather than globally exploring the entire Pareto front at once. In this context, preference learning operates at the subproblem level. We acknowledge, however, that sampled solutions may occasionally be mutually non-dominated in the original objective space. To investigate the effect of this scenario, we conducted an additional experiment where preference signals for such non-dominated pairs were set to 0, effectively treating them as equivalent. The results, shown in Table K, indicate that this modification (NDS) significantly degrades model performance across all Bi-TSP instances.
>
> This observation suggests that introducing indifference signals for non-dominated pairs hinders the learning process. One reason is that it reduces the amount of effective preference supervision per subproblem, especially in early training stages when many solutions are of similar quality. In contrast, using scalarized values provides a consistent and differentiable training signal aligned with the decomposition objective. Therefore, although scalarized comparisons may not reflect global Pareto dominance in every case, they remain well-justified and necessary within the decomposition-based learning paradigm, and do not appear to bias the overall Pareto front approximation based on our empirical findings.
>
> **Table K: Perfomance on Bi-TSP instances when neutralizing non-dominated pairs.**
> | Problem Size | POCCO-W HV | NDS HV |
> |--------------|:----------:|:------:|
> | Bi-TSP20      | 0.6275     | 0.5516 |
> | Bi-TSP50      | 0.6411     | 0.5160 |
> | Bi-TSP100     | 0.7055     | 0.5601 |
> | Bi-TSP150     | 0.7033     | 0.5494 |
> | Bi-TSP200     | 0.7371     | 0.5809 |
>
> **Q1: How is CCO routing learned, and what role does the Top-k play?**
>
> We have provided an explanation of the routing mechanism in Appendix C.3 and elaborate on it below.
>
> Our CCO block employs a subproblem-level gating mechanism to dynamically route each context vector $h_c$ to a subset of experts from a pool consisting of 4 FF experts and 1 ID expert (thus $m=4$). Let $d$ be the hidden dimension and $W_G \in \mathbb{R}^{d \times (m+1)}$ denote the trainable gating weight matrix. Given a batch of context vectors $X = \left[ h_c^b \right]_{b=1}^B \in \mathbb{R}^{B \times d}$, where $B$ is the batch size, the gating network computes a score matrix: $H = X \cdot W_G \in \mathbb{R}^{B \times (m+1)}$, where  each element $H_b^j$ represents the affinity or preference (score) of subproblem b towards expert $j$.
>
> For each subproblem, the router selects the Top-k highest-scoring experts from the score vector $H_b^j$.  These selected experts are then activated and their outputs are weighted by a softmax-normalized version of their scores. In our POCCO with $k = 2$, each subproblem is routed to its two highest-scoring experts.
>
> This Top-k routing plays a critical role by reducing computation, as only $k$ experts are activated per subproblem (i.e., sparse activation widely used in MoE structure [8]). It also encourages expert specialization, since different subproblems (defined by context and weight vectors) tend to activate different subsets of experts. Additionally, it allows the inclusion of a parameter-free ID expert, which is often selected when minimal transformation is needed. This enables the router to learn when it is appropriate to leave a representation unchanged.
>
> We analyzed the impact of Top-k selection on both training dynamics and final performance in Figure 6 (Appendix J). POCCO-W with Top-k = 2 achieves the fastest convergence and the highest final HV compared to Top-k = 1 and Top-k = 3. To further support this observation, Table I presents a detailed ablation study, where POCCO-W (Top-k = 2) consistently outperforms other settings across all tested problem sizes.
>
> **Table I: HV of dfferent Top-k on Bi-TSP instances.**
> | Problem Size | Top-k = 1 | Top-k = 2 | Top-k = 3 |
> |:------------:|:---------:|:---------:|:---------:|
> | Bi-TSP20  | 0.6274 | **0.6275** | **0.6275** |
> | Bi-TSP50  | 0.6404 | **0.6411** | 0.6409 |
> | Bi-TSP100 | 0.7046 | **0.7055** | 0.7050 |
> | Bi-TSP150 | 0.7020 | **0.7033** | 0.7025 |
> | Bi-TSP200 | 0.7354 | **0.7371** | 0.7364 |
>
>
> **Q3: What are the contributions of the CCO and PL to the overall speedup?**
>
> We would like to clarify that the proposed CCO block is not designed to improve computational efficiency, rather to enhance the model’s representational flexibility. In fact, as it introduces additional parameters through the router and FF experts, it incurs moderate overhead in both model size and runtime, as quantified in Tables 1-2 and Table 8.
>
> The efficiency gain we refer to in the paper primarily stems from our preference-driven training strategy and specifically refers to the model’s fast convergence during training. As shown in Figure 4(a) and Table L, our method achieves nearly 2× faster convergence compared to RL-based baselines under the same training conditions. This acceleration reduces the total wall-clock time required for training to reach high-quality solutions, which we consider a practical efficiency benefit during the learning phase.
>
> We acknowledge that the current wording in the paper may have caused confusion by implying overall computational efficiency. We are committed to eliminating any ambiguity in the final version.
>
> **Table L: HV of Different Training Algorithm at Each Epoch**
> | Method      | Epoch 1 | Epoch 50 | Epoch 100 | Epoch 150 | Epoch 200 |
> |-------------|:-------:|:-------:|:--------:|:--------:|:--------:|
> | POCCO-W (RL)| 0.6838  | 0.7009  | 0.7023   | 0.7032   | 0.7046   |
> | POCCO-W (PL)| 0.6884  | 0.7017  | 0.7037   | 0.7043   | 0.7055   |
>
> [1] Pareto set learning for neural multi-objective combinatorial optimization, in ICLR 2022.
>
> [2] Neural multi-objective combinatorial optimization with diversity enhancement, in NeurIPS 2023
>
> [3] Conditional neural heuristic for multiobjective vehicle routing problems, TNNLS, 2024.
>
> [4] Collaborative deep reinforcement learning for solving multi-objective vehicle routing problems, in AAMAS 2024.
>
> [5] Rethinking neural multi-objective combinatorial optimization via neat weight embedding, in ICLR 2025.
>
> [6] Towards understanding the mixture-of-experts layer in deep learning, in NeurIPS, 2022.
>
> [7] Statistical perspective of top-k sparse softmax gating mixture of experts, in ICLR 2024.
>
> [8] MVMoE: Multi‑Task Vehicle Routing Solver with Mixture‑of‑Experts, in ICML 2024.

---

> > ### Comment · Reviewer_CeFA · 2025-08-04
> >
> > I would like to thank the authors for their efforts to address my comments. My concerns have all been addressed. I will update the score accordingly.

---

> > > ### Author Response · Authors · 2025-08-04
> > >
> > > Thank you for your positive feedback and for updating the score. We sincerely appreciate your insightful and constructive comments. Your engagement has significantly contributed to improving the quality of our work.

---

### Official Review · Reviewer_ZHcK · 2025-07-02

**Clarity:** 3
**Significance:** 2
**Originality:** 2
**Rating:** 4
**Confidence:** 3

**Summary:**

The paper introduces POCCO (Preference-driven multi-Objective Combinatorial optimization with Conditional COmputation), a framework that can be wrapped around existing neural solvers for multi-objective combinatorial optimization problems (MOCOPs).
In particular, POCCO adds two orthogonal components:
1. Conditional Computation (CCO block): a sparse MoE / Mixture-of-Depth module inserted in the decoder that routes each scalarised sub-problem to a small, specialized subset of experts (or an identity path), efficiently scaling model capacity.
2. Preference-driven Learning (PL): Instead of REINFORCE with scalar rewards, the policy is trained on pairwise “win/lose” comparisons via a Bradley-Terry likelihood, using normalized log-likelihood as an implicit reward.

Integrated into two SOTA baselines (CNH and WE-CA), the resulting POCCO-C and POCCO-W outperform all neural and classical baselines on four benchmarks (Bi-/Tri-TSP, Bi-CVRP, Bi-KP) and show better OOD size generalization. The authors also discuss remaining limitations, such as scaling to real-world constraints and very large instances.

**Questions:**

## Questions

1. How does performance vary with Top-k and the number of experts? Could very large expert pools cause over-fragmentation or collapse?
2. Have you compared BT-likelihood with more recent objectives such as DPO?
3. How would CCO interact with the constraint-handling modules suggested for future work? Could routing specialise not just by weight vector but by constraint pattern?

## Possible typos

- l.169-170 – “it combines of a mixture-of-experts…” → “combines a mixture-of-experts”

**Ethical Concerns:**

["NO or VERY MINOR ethics concerns only"]

**Final Justification:**

I have updated my original score following the provided theoretical grounding and formulation of the PL objective and beginning of gradient variance analysis.

**Limitations:**

yes

**Paper Formatting Concerns:**

No formatting issues

**Quality:**

2

**Strengths And Weaknesses:**

## Strengths

- Novelty: Simple, general mechanisms that can be “plugged in” to many existing models.
- Experiments: 4 benchmarks × 3 sizes, 3 metric types, ablations on CCO variants and hyper-parameters, OOD tests.
- Training efficiency: PL halves convergence time vs. REINFORCE.
- Generalizability: Single model handles multiple sizes, beats multi-model baselines, and generalises to KroAB100-200 instances.
- Paper clarity: Background and motivation sections relate CCO + PL to known MoE and preference-learning literature.

## Weaknesses

- Theoretical grounding: No formal analysis of convergence or expressivity. All results rely solely on empirical evidence (the authors acknowledge no new theory).
- Experimental scope: Instances top out at 100 nodes for TSP/CVRP or 200 items for KP. Real-world constraints are left to future work.
- Missing ablations: CCO adds parameters, yet the paper does not quantify the overhead of expert routing.
- Writing: Several minor typos and grammatical errors that slightly hinder readability.

---

> ### Author Rebuttal · Authors · 2025-07-31
>
> Thank you for your insightful comments.
>
> **W1: Lacks theoretical analysis on convergence or expressivity.**
>
> Due to character limits, please refer to our response to Reviewer CeFA–W1 for details on W1.
>
> **W2: Limited problem scale; real-world constraints not addressed.**
>
> We mainly follow prior work [1-3], which primarily evaluates methods on MOTSP/MOCVRP with 20/50/100 nodes and MOKP with 50/100/200 items.
>
> To evaluate the out-of-distribution generalization ability of our method, we report additional results on larger Bi-TSP instances (150 and 200 nodes) as well as benchmark instances (KroAB150 and KroAB200) in Appendix H and I, respectively. Furthermore, to demonstrate scalability, we include new experimental results on even larger problems including Bi-TSP with 300/500/1000 nodes and MOKP with 300/500/1000 items. These results are summarized in Table D and Table E. As shown, POCCO-W consistently outperforms the baseline WE-CA in most large-scale cases.
>
> Following the reviewer’s suggestion to address the applicability of our method under more complex constraints, we conducted an additional experiment on the multi-objective capacitated vehicle routing problem with time windows (MOCVRPTW), as summarized in Table F. In the MOCVRPTW, each customer node $v_i$ is associated with a time window $[e_i, l_i]$ and a service time $s_i$[4]. A vehicle must start serving customer $v_i$ in the time slot from $e_i$ to $l_i$.  If the vehicle arrives earlier than $e_i$, it has to wait until $e_i$. All vehicles must return to the depot $v_0$ no later than $l_0$. This study considers three objectives: minimizing the total travel distance, minimizing the number of routes, and minimizing the average route length per customer served. As shown in Table F, POCCO-W consistently outperforms WE-CA on MOCVRPTW tasks as well.
>
> **Table E: HV on large Bi-TSP instances.**
> | Problem Size | WE-CA HV | POCCO-W HV |
> |--------------|:--------:|:----------:|
> | Bi-TSP300     | 0.7441   | **0.7458**     |
> | Bi-TSP500     | 0.7476   | **0.7592**     |
> | Bi-TSP1000    | 0.7186   | **0.7400**     |
>
> **Table F: HV on large MOKP instances.**
> | Problem Size | WE-CA HV | POCCO-W HV |
> |--------------|:--------:|:----------:|
> | Bi-KP300      | **0.6000**   | 0.5947     |
> | Bi-KP500      | 0.4244   | **0.5658**     |
> | Bi-KP1000     | 0.2439   | **0.8408**     |
>
> **Table G: HV on MOCVRPTW instances.**
> | Problem Size | WE-CA HV | POCCO-W HV |
> |--------------|:--------:|:----------:|
> | MOCVRPTW20    | 0.5465   | **0.5496** |
> | MOCVRPTW50    | 0.5499   | **0.5576** |
> | MOCVRPTW100   | 0.5331   | **0.5375** |
>
> **W3: Missing ablation on CCO’s routing overhead.**
>
> We have quantified the overhead introduced by the CCO block in Table 8 (Appendix L), which compares the total number of parameters across various neural MOCOP solvers. Our CCO block, designed to learn a diverse ensemble of policies, adds additional parameters yet achieves superior performance over prior methods. Specifically, the additional parameters in POCCO-W primarily come from four FF experts (totaling 526,848 parameters) and a lightweight router module (1,280 parameters). It is worth noting that during inference, each subproblem activates only two experts, one of which may be the parameter-free ID expert. This design helps keep the computational overhead manageable.
>
> To further address your concern, we conducted an additional set of experiments comparing several neural MOCOP solvers in terms of model size, training time, GPU memory usage (on Bi-TSP50), and solution quality (HV). As shown in Table H, while POCCO-W has moderately higher overhead than CNH and WE-CA, it achieves better solution quality. We believe this represents a reasonable trade-off between model complexity and performance.
>
> **Table H: Overhead of Different Neural MOCOP solvers.**
> | Method    | Parameters | Training Time| GPU Usage (Bi-TSP50)|HV (Bi-TSP50)|
> |-----------|:-----------:|:-------------:|:-----------:|:-----------:|
> | CNH       | 1.63M       | 17h |1077 MB | 0.6387 |
>  WE-CA     | 1.47M       | 11h |1051 MB | 0.6392 |
> | POCCO-W   | 2.00M   | 36h |1849 MB  | 0.6411 |
>
> **W4: Minor writing issues.**
>
> We acknowledge minor typos and grammatical issues in the manuscript and will thoroughly revise the paper to improve clarity and readability.
>
> **Q1:  How do Top-k and expert count affect performance? Risk of collapse?**
>
> We analyzed the impact of Top-k selection on both training dynamics and final performance in Figure 6 (Appendix J). POCCO-W with Top-k = 2 achieves the fastest convergence and the highest final HV compared to Top-k = 1 and Top-k = 3. To further support this observation, Table I presents a detailed ablation study, where POCCO-W (Top-k = 2) consistently outperforms other settings across all tested problem sizes.
>
> We have added an ablation study to analyze the impact of the number of experts in Table A (due to character limits, please refer to our response to Reviewer ETVh–W2 for details on Table A). Specifically, POCCO-W (5E) refers to the original model presented in the paper, which includes 4 feedforward (FF) experts and 1 parameter-free identity (ID) expert. We additionally evaluate four variants:
> * POCCO-W (3E): 2 FF + 1 ID;
> * POCCO-W (9E): 8 FF + 1 ID;
> * POCCO-W (9E_2D): 8 FF + 1 ID, trained on twice the amount of data;
> * POCCO-W (17E): 16 FF + 1 ID.
>
> All variants are trained under the same settings as POCCO-W (5E) for a fair comparison, except for POCCO-W (9E_2D), which is trained on more data.
>
> As shown in Table A, all POCCO-W variants outperform the backbone model WE-CA, confirming the effectiveness of the expert-based architecture. Moreover, POCCO-W (3E) and POCCO-W (5E) achieve better overall performance than POCCO-W (9E) and POCCO-W (17E), the latter of which require additional data scaling to realize performance gains. To strike a better balance between computational cost and solution quality, we select POCCO-W (5E) as our default model.
>
> Due to computational resource constraints and rebuttal timeframe limitations, we did not test even larger expert pools (e.g., 25 or 49 experts). Nevertheless, POCCO-W (17E) still performs robustly, showing no collapse. We will further explore and provide deeper analyses with larger expert pools in the final manuscript.
>
> **Table I: HV of dfferent Top-k on Bi-TSP instances.**
> | Problem Size | Top-k = 1 | Top-k = 2 | Top-k = 3 |
> |:------------:|:---------:|:---------:|:---------:|
> | Bi-TSP20  | 0.6274 | **0.6275** | **0.6275** |
> | Bi-TSP50  | 0.6404 | **0.6411** | 0.6409 |
> | Bi-TSP100 | 0.7046 | **0.7055** | 0.7050 |
> | Bi-TSP150 | 0.7020 | **0.7033** | 0.7025 |
> | Bi-TSP200 | 0.7354 | **0.7371** | 0.7364 |
>
> **Q2: Comparison with DPO?**
>
> We have added a comparative experiment between our preference learning (PL) method and the recent DPO objective, as shown in Table H. Since DPO requires two models, i.e., a policy model and a reference model, we use the same architecture and initialization for both, with the reference model updated from the previous epoch. This setup significantly increases training time for DPO (122 h vs. 36 h for POCCO-W).
>
> As shown in Table H, while DPO achieves slightly better performance on the smallest instance (MOTSP20), it is consistently outperformed by POCCO-W on larger instances (MOTSP50–200). These results suggest that our PL method offers a more favorable trade-off in terms of both training efficiency and solution quality, especially for large-scale problems.
>
> **Table J: HV of different PL methods on Bi-TSP instances.**
> | Problem Size | POCCO-W  | DPO   |
> |--------------|:----------:|:-------:|
> | Bi-TSP20      | 0.6275     | **0.6276**  |
> | Bi-TSP50      | **0.6411**     | 0.6400  |
> | Bi-TSP100     | **0.7055**     | 0.7030  |
> | Bi-TSP150     | **0.7033**     | 0.6998  |
> | Bi-TSP200     | **0.7371**     | 0.7331  |
>
> **Q3: Can CCO routing adapt to constraint patterns?**
>
> Our CCO module between the decoder’s MHA and the compatibility layer is orthogonal to the constraint-handling mechanism, which is primarily realized via masking schemes in the MHA and compatibility modules. In future work, we envision two possible directions where CCO could interact with constraint-handling modules:
> * **Single-task with hard constraints:** MOCOPs with hard constraints (e.g., TSPTW) are challenging, as identifying valid masking patterns is itself an NP-hard problem. In such cases, we could use an auxiliary decoder to learn to predict feasible masking patterns [5], while the main decoder with the CCO learns the solution policy. Through joint training or mutual distillation, the expert routing in CCO can specialize not just by weight vector, but also implicitly by constraint structure. This interaction encourages experts to develop sub-policies aligned with recurring constraint motifs.
> * **Multi-task generalization with varying constraint patterns:** For multi-task settings where the model needs to generalize across multiple MOCOPs with heterogeneous constraints (e.g., TSP with capacity, VRP with time windows, KP with budget limits), we can extend the input to the CCO router by concatenating a binary constraint indicator vector that encodes the presence or type of constraints for each task [4]. This approach allows experts to specialize in sub-distributions of tasks sharing similar constraint structures.
>
> In both settings, the CCO module functions as a flexible capacity allocator within the decoder, and its interaction with constraint-handling modules can be enhanced either implicitly or explicitly.
>
> [1] Pareto set learning for neural multi-objective combinatorial optimization, in ICLR 2022.
>
> [2] Conditional neural heuristic for multiobjective vehicle routing problems, TNNLS, 2024.
>
> [3] Rethinking neural multi-objective combinatorial optimization via neat weight embedding, in ICLR 2025.
>
> [4] MVMoE: Multi‑Task Vehicle Routing Solver with Mixture‑of‑Experts, in ICML 2024.
>
> [5] Learning to handle complex constraints for vehicle routing problems, in NeurIPS 2024.

---

> > ### Comment · Reviewer_ZHcK · 2025-08-05
> >
> > Thank you for the detailed and comprehensive rebuttal. I appreciate the new large-scale results, MOCVRPTW evaluation, and ablation studies on Top-k and expert count, which effectively address concerns around scalability and routing overhead. The clarification on constraint-handling potential and comparison with DPO further strengthen the paper. While theoretical analysis remains limited, the added empirical evidence improves confidence in the method’s generality and effectiveness. I maintain my original borderline reject rating due to the absence of formal grounding and modest innovation, but acknowledge the revised submission is much stronger and close to the acceptance threshold.

---

> > > ### Author Response · Authors · 2025-08-07
> > > **Further Response (1/2)**
> > >
> > > Thanks for your further feedback. We hope our response below addresses your comments.
> > >
> > > ## **Theoretical Analysis**
> > >
> > > Most prior work in neural-based MOCOP has been predominantly empirical, and providing a rigorous theoretical treatment is non-trivial within the rebuttal timeline. Nevertheless, in response to your comment, we have conducted a preliminary theoretical analysis. Specifically, we derive the loss function and gradient for our PL objective and contrast it with the REINFORCE gradient. Our theoretical (and empirical) analyses demonstrate that **PL yields significantly lower gradient variance, which contributes to faster and more stable convergence.** We will include a detailed theoretical discussion in the final version.
> > >
> > > ### **Formulation of Loss Function**
> > >
> > > Different from the RL, our PL method exploits the preference relations between generated solutions according to their objective. The explicit preference $f^* (\pi|\mathcal{G}, \lambda)$ is defined as the negation of the scalarized objective. We contruct a preference pair, denoted as $(\pi^w, \pi^\ell)$, along with a binary preference label $y$, where $y = 1$ if $\pi^{w} \lessdot \pi^{l}$ (i.e., $f^* (\pi^w|\mathcal{G}, \lambda) > f^* (\pi^\ell|\mathcal{G}, \lambda)$), and $y = 0$ otherwise.
> > >
> > > For the policy $p_\theta(\pi|\mathcal{G}, \lambda)$ used to construct solution $\pi$, its implicit preference is defined as the average log-likelihood [1]: $f_\theta (\pi |\mathcal{G}, \lambda)= \frac{1}{|\pi|} \log p_{\theta} (\pi| \mathcal{G}, \lambda).$ For a preference pair $(\pi^w, \pi^\ell)$, the preference distribution is modeled using the Bradley-Terry (BT) ranking objective [2] and implicit preferences:
> > >
> > > $$
> > > g_{\theta} \bigl (\pi^{w} \lessdot \pi^{\ell}\mid\mathcal{G},\lambda \bigr)=
> > > \sigma \bigl( \beta [f_{\theta}(\pi^{w}\mid\mathcal{G},\lambda) - f_{\theta}(\pi^{\ell}\mid\mathcal{G},\lambda)]\bigr),
> > > $$
> > >
> > > where $\sigma(\cdot)$ is the sigmoid function, and $\beta>0$ is a fixed temperature that controls the sharpness with which the model distinguishes between unequal rewards. By maximizing the log-likelihood of $g_{\theta} \bigl (\pi^{w} \lessdot \pi^{\ell}\mid\mathcal{G},\lambda \bigr)$, the model is encouraged to assign higher probabilities to preferred solutions $\pi^w$ compared with less preferred solutions $y^\ell$. We can derive the PL loss function:
> > >
> > > $$
> > > \mathcal{L}_ {PL} (\theta|p_\theta, \mathcal{G},\lambda, \pi^w, \pi^l) = - y \log \sigma \bigl( \beta [\frac{\log p_\theta(\pi^w|\mathcal{G},\lambda)}{|\pi^w|}-\frac{\log p_\theta(\pi^l|\mathcal{G},\lambda)}{|\pi^l|}\bigr]),
> > > $$
> > >
> > > PL directly distinguishes optimization signals based on exact preferences, and the strength of the optimization signal correlates with the difference in log-likelihood, requiring the model to maximize the probabilities gap between $\pi^w$ and $\pi^\ell$.
> > >
> > >
> > > ### **Gradient Analysis**
> > >
> > > Let $z$ denote the argument of the sigmoid function:
> > > $$
> > > z=\beta [\frac{\log p_\theta(\pi^w|\mathcal{G},\lambda)}{|\pi^w|}-\frac{\log p_\theta(\pi^l|\mathcal{G},\lambda)}{|\pi^l|}\bigr].
> > > $$
> > >
> > > The gradient of $\mathcal{L}_ {PL}$ with respect to $\theta$ is:
> > > $$
> > > \nabla_\theta \mathcal{L}_ {PL} = \frac{\partial \mathcal{L}_ {PL}}{\partial z} \cdot \nabla_\theta z.
> > > $$
> > >
> > > Derivative of $−y \text{log} \sigma(z)$ becomes:
> > > $$
> > > \frac{\partial \mathcal{L}_ {PL}}{\partial z} = - y(1 - \sigma(z)).
> > > $$
> > >
> > > Gradient of $z$ with respect to $\theta$:
> > > $$
> > > \nabla_\theta z = \beta [\frac{1}{|\pi^w|} \nabla_\theta \log p_\theta(\pi^w|\mathcal{G},\lambda) - \frac{1}{|\pi^\ell|} \nabla_\theta \log p_\theta(\pi^\ell | \mathcal{G},\lambda)].
> > > $$
> > >
> > > Combining these, the total gradient becomes:
> > > $$
> > > \nabla_\theta \mathcal{L}_ {PL}= - y \beta(1 - \sigma(z))[\frac{1}{|\pi^w|} \nabla_\theta \log p_\theta(\pi^w|\mathcal{G},\lambda) - \frac{1}{|\pi^\ell|} \nabla_\theta \log p_\theta(\pi^\ell | \mathcal{G},\lambda)].
> > > $$
> > >
> > > The gradient increases the likelihood of $\pi^w$ and decreases the likelihood of $\pi^\ell$. For comparison, we analyze the loss function in the REINFORCE algorithm here:
> > > $$
> > > \mathcal{L}_ {RL} (\pi|\mathcal{G}, \lambda)=-(\mathcal{R}(\pi)-b) \text{log} p_{\theta} (\pi|\mathcal{G},\lambda),
> > > $$
> > >
> > > where $b$ is a baseline to distinguish positive or negative optimization signals for each solution $\pi$. The gradient of the REINFORCE algorithm is:
> > >
> > > $$
> > > \nabla_\theta \mathcal{L}_ {RL} = - (\mathcal{R}(\pi)-b) \nabla_\theta \text{log} p_\theta (\pi|\mathcal{G},\lambda).
> > > $$
> > >
> > > REINFORCE relies on **absolute** returns $\mathcal{R}(\pi)$, whose large variance propagates directly to the gradient. In contrast, $\mathcal{L}_{\mathrm{PL}}$ uses **relative** returns inside a sigmoid, yielding the difference of two normalized log-likelihoods.  This pairwise, centered signal reduces gradient variance and produces smoother, more stable updates, leading to faster and more reliable convergence. This theoretical insight is further supported by our empirical analyses (see our response to `Reviewer CeFA W1`).

---

> > ### Comment · Reviewer_ZHcK · 2025-08-05
> >
> > Thank you for the detailed and comprehensive rebuttal. I appreciate the new large-scale results, MOCVRPTW evaluation, and ablation studies on Top-k and expert count, which effectively address concerns around scalability and routing overhead. The clarification on constraint-handling potential and comparison with DPO further strengthen the paper. While theoretical analysis remains limited, the added empirical evidence improves confidence in the method’s generality and effectiveness. I maintain my original borderline reject rating due to the absence of formal grounding and modest innovation, but acknowledge the revised submission is much stronger and close to the acceptance threshold.

---

> ### Author Response · Authors · 2025-08-07
> **Further Response (2/2)**
>
> ## **Innovation**
>
> Regarding innovation, we would like to highlight that our work introduces POCCO, **a general, plug-and-play framework** that enhances neural MOCOP methods through two complementary and, to the best of our knowledge, novel mechanisms:
>
> * First, POCCO incorporates a context-aware routing module into the decoder, enabling subproblems to dynamically select computation paths (i.e., model structures) based on their individual contexts. This adaptively scales model capacity and improves representation learning.
> * Second, POCCO replaces scalarized rewards with a pairwise preference learning objective, which guides the policy search toward more preferred solutions. This formulation facilitates more effective exploration of promising regions in the solution space and leads to faster convergence toward higher-quality solutions, as supported by our theoretical and empirical analyses.
>
> These contributions, which have not been explored in prior MOCOP literature to our knowledge, offer a meaningful step forward in advancing neural methods for MOCOP. The novelty and value of our work have also been positively acknowledged by four other reviewers in their originality ratings. While we understand that assessments of novelty can be subjective, we believe our contributions offer a solid and practical advancement for the MOCOP community.
>
> Thank you again for your valuable time and constructive feedback. We would be happy to clarify or discuss further if anything remains unclear.
>
> ```
> [1] Simpo: Simple preference optimization with a reference-free reward, in NeurIPS, 2024.
> [2] Rank analysis of incomplete block designs: I. the method of paired comparisons. Biometrika, 1952.
> ```

---

> > ### Comment · Reviewer_ZHcK · 2025-08-08
> >
> > I thank the authors for fully deriving the PL objective and offering the above variance analysis. In my opinion, this derivation increases clarity and brings more grounding to the work, which was my main original criticism. I will therefore update the score accordingly, and I will recommend that you include this theoretical grounding in the manuscript.
> >
> > PS: There probably is a need to include expectations over sampled $\pi$ in the loss and gradient analysis.
> >
> > Thank you for the additional work and for addressing my concerns!
> > Best

---

> > > ### Author Response · Authors · 2025-08-09
> > >
> > > Thank you for the positive feedback and for recommending a score update. We will incorporate the theoretical analysis into the manuscript.
> > >
> > > Regarding your PS: we agree and will revise the manuscript accordingly to make the expectations explicit in the loss and gradient analysis.
> > >
> > > We sincerely appreciate your careful reading and constructive suggestions. Your engagement throughout the review process has greatly improved the readability and methodological strength of our work.

---

### Official Review · Reviewer_ETVh · 2025-07-03

**Clarity:** 4
**Significance:** 3
**Originality:** 3
**Rating:** 4
**Confidence:** 3

**Summary:**

Current methods for multi-objective combinatorial optimization problems (MOCOPs) often rely on a single neural network to handle all subproblems with different weight vectors, resulting in a limited model capacity for diverse subproblems. On the other hand, current methods usually use REINFORCE and use scalarized objective values as the reward signals, introducing high variance. To address these issues, the paper introduces a novel method called POCCO, which increases model capacity by conditional computational (CCO) block and utilize preference-based reward for a better convergence. Experiments across several classic MOCOP benchmarks show that POCCO outperforms all the learning-based methods, with an increased but still acceptable computational time cost.

**Questions:**

- Could you provide a comparison on the resource usage (e.g., GPU memory usage) of the methods? This would provide a more complete understanding of the practical trade-offs.
- Can POCCO be used in the tasks where preference among the objective cannot be presented as a linear weighted sum, but only a more complex combination (e.g. Tchebycheff scalarization)?

I am willing to change my rating if all the concerns and questions is answered or addressed.

**Ethical Concerns:**

["NO or VERY MINOR ethics concerns only"]

**Final Justification:**

The authors have addressed most of my concerns. I am planning to keep my rating unchanged.

**Limitations:**

The CCO block may require more computational cost (including time and resource usage) compared to other learned-based methods.

**Paper Formatting Concerns:**

None.

**Quality:**

3

**Strengths And Weaknesses:**

Strengths

- The paper introduces both a new architecture and a new training method for MOCOPs, which are conceptually sound and well-motivated for MOCOPs.
- In the experiment, the proposed method POCCO outperforms all the learning-based methods in terms of hyper-volume and gap. The time usage is longer but acceptable. The experiment is generally adequate.
- The paper is clearly written, well-organized, and easy to follow.

Weaknesses

- The CCO block may introduce more computational overhead (including time and resource usage) compared to other learned-based methods. The longer computational time is evident in the reported experimental results.
- The influence of the number of experts is not analyzed. It might be critical to the performance.
- The source code is not yet available in the supplementary material, which may hinder the evaluation on reproducibility.

---

> ### Author Rebuttal · Authors · 2025-07-31
>
> We genuinely value and appreciate the time and effort the reviewer dedicated to evaluating our paper and offering insightful feedback. Here are our detailed responses to your comments, where W denotes Weakness (W1-W3) and Q denotes Question (Q1-Q2).
>
> **W1: The CCO block may introduce more computational overhead (including time and resource usage) compared to other learned-based methods.**
>
> We acknowledge that the CCO block introduces additional computational overhead due to the extra parameters added on top of the backbone model. While this results in a moderate increase in runtime (as shown in Tables 1-2) and resource usage (as noted in our response to `Q1`), the improvement in solution quality is substantial. As demonstrated in our ablation study, the CCO block plays a key role in achieving these performance gains, justifying its inclusion in the overall design. We believe this trade-off between efficiency and performance is reasonable, especially considering the gains over other learning-based methods.
>
> **W2: The influence of the number of experts is not analyzed.**
>
> Thanks for your comment. We have added an ablation study to analyze the impact of the number of experts in Table A. Specifically, POCCO-W (5E) refers to the original model presented in the paper, which includes 4 feedforward (FF) experts and 1 parameter-free identity (ID) expert. We additionally evaluate four variants:
> * POCCO-W (3E): 2 FF + 1 ID;
> * POCCO-W (9E): 8 FF + 1 ID;
> * POCCO-W (9E_2D): 8 FF + 1 ID, trained on twice the amount of data;
> * POCCO-W (17E): 16 FF + 1 ID.
>
> All variants are trained under the same settings as POCCO-W (5E) for a fair comparison, except for POCCO-W (9E_2D), which is trained on more data.
>
> As shown in Table A, all POCCO-W variants outperform the backbone model WE-CA, confirming the effectiveness of the expert-based architecture. Moreover, POCCO-W (3E) and POCCO-W (5E) achieve better overall performance than POCCO-W (9E) and POCCO-W (17E), the latter of which require additional data scaling to realize performance gains. To strike a better balance between computational cost and solution quality, we select POCCO-W (5E) as our default model.
>
> **Table A: HV on Bi-TSP Instances.**
> | Problem Size | WE-CA | POCCO-W (3E)| POCCO-W (5E)  | Exp_num (9E) |Exp_num (9E_2D) |Exp_num (17E) |
> |:-------------|:--------:|:-----------------:|:------------------------:|:-----------------:|:-----------------:|:-----------------:|
> | Bi-TSP20  | 0.6270 | **0.6275** | **0.6275** | **0.6275** |**0.6275**|**0.6275** |
> | Bi-TSP50  | 0.6392 | 0.6410 | **0.6411** | 0.6408 | **0.6411** | 0.6408 |
> | Bi-TSP100 | 0.7034 | **0.7057** | 0.7055 | 0.7050 |   0.7053   | 0.7050 |
> | Bi-TSP150 | 0.7008 | **0.7035** | 0.7033 | 0.7026 |   0.7031 | 0.7027 |
> | Bo-TSP200 | 0.7346 | 0.7363 | **0.7371** | 0.7364 |   0.7370| 0.7361 |
>
>
> **W3: The source code is not yet available.**
>
> We are committed to publicly releasing our code and dataset upon acceptance. Due to NeurIPS 2025 rebuttal policy, which prohibits the inclusion of external links, we were unable to share our source code via an anonymous repository. However, we will ensure full code and data availability to support reproducibility once the paper is published.
>
>
> **Q1: Could you provide a comparison on the resource usage (e.g., GPU memory usage) of the methods? This would provide a more complete understanding of the practical trade-offs.**
>
> Following the reviewer's suggestion, we present a comparison of GPU memory usage between POCCO-W (ours), the backbone model (WE-CA), and CNH in Table B. All models are trained as unified models across problem instances with size $n =50$. As shown in Table B, POCCO-W incurs higher GPU memory usage, approximately doubling that of the baselines WE-CA and CNH. However, this increased resource cost is accompanied by significant improvements in solution quality, as demonstrated in our experimental results. We will include a more detailed comparison in the final version of the paper to provide a clearer understanding of the trade-offs involved.
>
> **Table B: Comparison of GPU Memory Usage.**
>
> | Problem Size| WE-CA |CNH |POCCO-W |
> |-----|-----|-----|-----|
> | Bi-TSP50  |  1051 MB |1077 MB | 1849 MB|
> | MOCVRP50  |  1790 MB |1501 MB | 4522 MB|
> | Bi-KP50  |  916 MB |916 MB | 2417 MB|
> | Tri-TSP50  |  1052 MB |1077 MB|  2383 MB|
>
>
> **Q2: Can POCCO be used in the tasks where preference among the objective cannot be presented as a linear weighted sum, but only a more complex combination (e.g. Tchebycheff scalarization)?**
>
> Intuitively, our POCCO framework is applicable to any MOCOP where the objectives can be scalarized using decomposition-based techniques, including weighted-sum (WS), Tchebycheff (TCH), and penalty-based boundary intersection (PBI). This is because the preference signal between two solutions is determined solely by the ordering of their scalarized objective values, regardless of the specific scalarization method used.
>
> To empirically demonstrate this generality, we added an experiment comparing POCCO-W trained with WS and TCH scalarization methods. As shown in Table C, WS consistently outperforms TCH on Bi-TSP50 to Bi-TSP200, indicating that WS offers more robust performance across different problem sizes. This observation is also consistent with findings in [1], which report that WS—despite its simplicity—often outperforms TCH in the studied MOCOP setting. Due to the limited rebuttal period, we will include the detailed results for WE-CA (TCH) in the revised paper. In general, however, WE-CA (TCH) performs worse than WE-CA (WS), which achieves a score of 0.6392, and is further outperformed by POCCO-W (TCH), which achieves 0.6395 on Bi-TSP50.
>
> **Table C: HV of Different Decomposition Approaches on Bi-TSP Instances.**
> | Problem Size | WE-CA (WS) | POCCO-W (WS) | POCCO-W (TCH) |
> | :----------: | :--------: | :----------: | :-----------: |
> |   Bi-TSP50   |   0.6392   |    0.6411    |    0.6395     |
> |  Bi-TSP100   |   0.7034   |    0.7055    |    0.7031     |
> |  Bi-TSP150   |   0.7008   |    0.7033    |    0.6998     |
> |  Bi-TSP200   |   0.7346   |    0.7371    |    0.7329     |
>
> [1] Rethinking neural multi-objective combinatorial optimization via neat weight embedding, in ICLR 2025.

---

> > ### Comment · Reviewer_ETVh · 2025-08-08
> >
> > Thank you for the detailed response, which addresses most of my concerns. I am planning to keep my positive rating unchanged.

---

> > > ### Author Response · Authors · 2025-08-09
> > >
> > > Thank you for your follow-up and for confirming that most of your concerns have been addressed. We appreciate your positive evaluation and are grateful for your support.

---

> ### Author Response · Authors · 2025-08-07
>
> Dear Reviewer `ETVh`,
>
> Thank you so much for your time and efforts in reviewing our paper!
>
> As the author-reviewer discussion period is extended for in-depth discussion, we sincerely hope there would be a chance for us to clarify our work with you! Please kindly let us know if you have any further questions or concerns.
>
> We are looking very forward to your reply. Thank you once again!
>
> Best,
> Authors

---

> ### Author Response · Authors · 2025-08-08
>
> Dear Reviewer `ETVh`,
>
> Thank you for the time and effort you have devoted to reviewing our paper.
>
> With **fewer than 24 hours** remaining in the discussion period, we would appreciate the opportunity to clarify any points that remain unclear. Please let us know if you have additional questions or concerns.
>
> We look forward to your reply. Thank you again!
>
> Best regards,
> Authors

---

### Decision · Program_Chairs · 2025-09-17

**Decision:**

Accept (poster)

**Comment:**

This study introduces a new architecture for multi-objective combinatorial optimization using deep reinforcement learning, featuring two key components: conditional computation and preference learning. The main contribution of this research is empirical evidence that the POCCO architecture outperforms all baseline models across several benchmark tests.

Overall, the paper has received mixed reviews, ranging from borderline acceptance to clear endorsement. Furthermore, the authors addressed both theoretical and experimental concerns effectively during the rebuttal phase. Therefore, I recommend acceptance of the paper.

In the revised version, it would be beneficial to include the improvements made during the rebuttal phase. Specifically, the related work section should be integrated into the main paper. Additionally, further experiments covering important aspects—such as additional baselines, the impact of the number of experts, top-k selection of experts, and real-world constraints—could be included either in the main paper or the appendix. Finally, the theoretical analysis should also be provided in the appendix.